# Detection of haplotype-dependent allele-specific DNA methylation in WGBS data

J. Abante [1,2✉], Y. Fang [3,4], A. P. Feinberg [3,4,5] & J. Goutsias [1,2✉]

In heterozygous genomes, allele-specific measurements can reveal biologically significant differences in DNA methylation between homologous alleles associated with local changes in genetic sequence. Current approaches for detecting such events from whole-genome bisulfite sequencing (WGBS) data perform statistically independent marginal analysis at individual cytosine-phosphate-guanine (CpG) sites, thus ignoring correlations in the methylation state, or carry-out a joint statistical analysis of methylation patterns at four CpG sites producing unreliable statistical evidence. Here, we employ the one-dimensional Ising model of statistical physics and develop a method for detecting allele-specific methylation (ASM) events within segments of DNA containing clusters of linked single-nucleotide polymorphisms (SNPs), called haplotypes. Comparisons with existing approaches using simulated and real WGBS data show that our method provides an improved fit to data, especially when considering large haplotypes. Importantly, the method employs robust hypothesis testing for detecting statistically significant imbalances in mean methylation level and methylation entropy, as well as for identifying haplotypes for which the genetic variant carries significant information about the methylation state. As such, our ASM analysis approach can potentially lead to biological discoveries with important implications for the genetics of complex human diseases.

[1] Whitaker Biomedical Engineering Institute, Johns Hopkins University, Baltimore, MD 21218, USA. [2] Department of Electrical & Computer Engineering, Johns Hopkins University, Baltimore, MD 21218, USA. [3] Center for Epigenetics, Johns Hopkins University School of Medicine, Baltimore, MD 21205, USA. [4] Department of Biomedical Engineering, Johns Hopkins University, Baltimore, MD 21205, USA. [5] Department of Medicine, Johns Hopkins University School of Medicine, Baltimore, MD 21205, USA. ✉email: jabante1@jhu.edu; goutsias@jhu.edu

In diploid mammalian genomes, homologous chromosomes may exhibit sequence-dependent imbalances in DNA methylation at potentially important genomic regions[1]. Identifying these regions and understanding the relationship between DNA methylation and sequence constitutes the focal interest of an active area of epigenetic research known as allele-specific methylation (ASM) analysis. In fact, ASM analysis underpins several reported associations between *cis*-genetic variation and DNA methylation imbalances in homologous chromosomes[1–5].

A type of ASM analysis is performed within specific DNA segments, known as haplotypes[1,3–6]. These regions contain heterozygous single-nucleotide polymorphisms (SNPs) that are responsible for genetic differences between the two homologous chromosomes in a given subject. Haplotype-dependent ASM (hap-ASM) analysis requires simultaneous observation of the DNA methylation state within each allele (genetic variant of a

haplotype) and the allele of origin. Heterozygous SNPs can be identified by whole-genome sequencing (WGS), whereas whole-genome bisulfite sequencing (WGBS) can be used to obtain observations of the DNA methylation state which are then aligned to a reference genome to determine their allele of origin.

Currently, there are two basic approaches to perform hap-ASM analysis (see Supplementary Methods, Section 1, for details). The first approach, which we refer to as the non-parametric independent (NPI) method, assumes that DNA methylation occurs in a statistically independent manner at individual cytosine-phosphate-guanine (CpG) sites, whereas the second approach, which we refer to as the non-parametric dependent (NPD) method, does not make use of such a strong assumption. The NPI method, previously employed for ASM analysis by Gertz et al.[7] and Fang et al.[8], is based on empirically estimating the marginal probability of methylation at each CpG site using WGBS reads of the methylation state only at the particular CpG site (see Fig. 1). At low levels of correlation between the methylation states at few contiguous CpG sites, the NPI method can perform satisfactorily well. However, this approach will not produce reliable results in the presence of high correlations, which are prevalent in WGBS data[9,10], especially when the region of interest contains a larger number of CpG sites. Moreover, the unrealistic assumption of statistical independence may lead to loss of specificity (true negative rate) and sensitivity (true positive rate)[11], which will seriously affect the method's statistical performance.

The NPD method, employed for ASM analysis by Onuchic et al.[12], is based on empirically estimating the joint probability of methylation at four contiguous CpG sites, known as epialleles, located around individual SNPs. This estimation can only be done using fully observed methylation reads of the epiallelic state (see Fig. 1), demanding a much higher coverage for reliable estimation and analysis than the one provided by currently available WGBS technologies. Insufficient coverage can lead to large uncertainty and low accuracy when estimating epiallelic probabilities using the NPD method, especially in areas of the genome that exhibit high methylation stochasticity (see ref. [10] and Supplementary Note, Section 3), which can seriously affect downstream statistical analysis. This problem is exacerbated when epialleles that contain more than four CpG sites are used in the analysis, due to the geometric growth of the number of epiallelic patterns associated with an increasing number of CpG sites. Consequently, the NPD method is not appropriate for hap-ASM analysis, which often requires estimation of joint methylation probabilities within genomic regions that contain more than four CpG sites.

In this work, we use concepts from statistical physics and information theory and develop a parametric approach for ASM analysis, based on the one-dimensional Ising model of statistical physics, which extends our earlier developments of potential energy landscape analysis of DNA methylation[10] to hap-ASM. By using simulated and real data, we show that this approach, which we refer to as the correlated potential energy landscape (CPEL) method, effectively addresses the previous limitations. This is achieved by jointly modeling methylation data over multiple CpG sites, by consistently providing accurate estimates of the joint probability distributions of methylation (PDM) at these CpG sites, and by constructing a reliable statistical approach for hap-ASM analysis that outperforms existing methods.

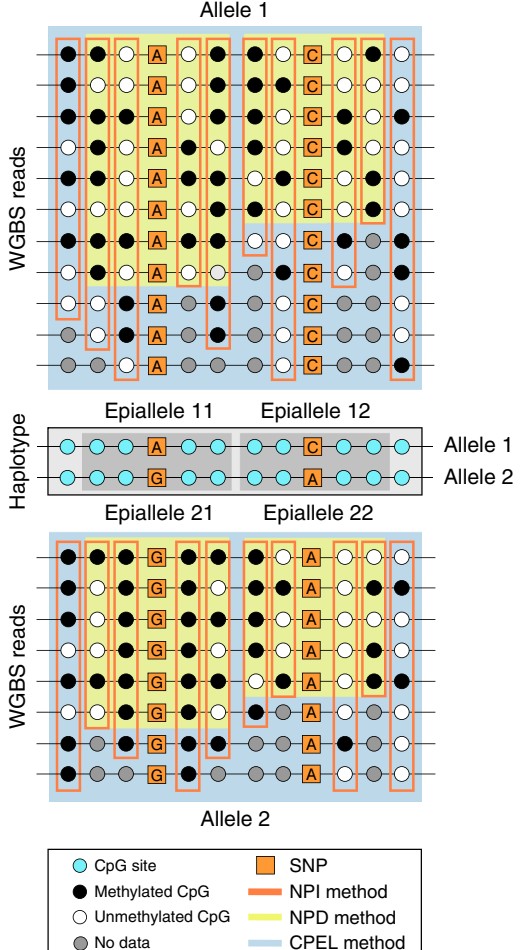

**Fig. 1 Comparison of hap-ASM analysis methods.** WGBS reads (rows) corresponding to each homologous allele of a haplotype are used by each method to estimate the probability distribution of methylation (PDM) within each allele. The NPI method provides an empirical estimate of the PDM at each CpG site using WGBS observations of its methylation state (orange rectangles). The NPD method provides an empirical estimate of the joint PDM within an epiallele, composed of a few CpG sites (usually 4) around an SNP, using only full WGBS observations of its methylation state (yellow rectangles). The CPEL method performs haplotype-dependent allele-specific methylation analysis by estimating the joint PDM at all CpG sites within a homologous allele using the entire set of WGBS data (blue rectangles), thus utilizing all available methylation information and allowing modeling of the cooperative effect of multiple SNPs on DNA methylation.

## Results

**Method overview.** To take into account the cooperative effect of SNP clusters on DNA methylation, proper ASM analysis must be performed at the haplotype level. The CPEL method provides a statistical approach to hap-ASM analysis that is capable of

evaluating imbalances in DNA methylation stochasticity between homologous alleles in a haplotype, properly identified from WGS and SNP data using read-based SNP phasing (see "Methods"), by leveraging concepts from statistical physics and information theory. It achieves this by modeling DNA methylation within a homologous allelic region $\mathcal{R}$ that contains $N$ CpG sites $n = 1, 2, \ldots, N$ using the $N \times 1$ random methylation state vector $\mathbf{X} = [X_1, X_2, \ldots, X_N]^{\mathrm{T}}$, where $X_n = 0$ if the $n$th CpG site is unmethylated and $X_n = 1$ if it is methylated. Subsequently, it partitions $\mathcal{R}$ into the minimum number of equally-sized non-overlapping allelic subregions $\mathcal{R}_k$, $k = 1, 2, \ldots, K$, such that the size of each subregion does not exceed 500 bp (see Supplementary Methods, Section 2, for details). It then characterizes $\mathbf{X}$ using the probabilities $p(\mathbf{x}) = \mathrm{Pr}[\mathbf{X} = \mathbf{x}]$, given by

$$p(\mathbf{x}) = \frac{1}{Z} \exp\{-U(\mathbf{x})\}, \text{ for every } \mathbf{x} \in \{0, 1\}^N, \quad (1)$$

where

$$U(\mathbf{x}) = -\sum_{k=1}^{K} \alpha_k \sum_{n \in \mathcal{N}_k} (2x_n - 1) - \beta \sum_{n=1}^{N-1} (2x_n - 1)(2x_{n+1} - 1) \quad (2)$$

is the potential energy of a methylation state $\mathbf{x}$, and

$$Z = \sum_{\mathbf{x}} \exp\{-U(\mathbf{x})\} \quad (3)$$

is a normalizing constant, known as the partition function, which is evaluated by the CPEL method in a computationally efficient manner (see Supplementary Methods, Sections 3 and 4, for details). In Eq. (2), $\mathcal{N}_k$ is the set of all CpG sites within the allelic subregion $\mathcal{R}_k$, whereas $\alpha_k$ is a parameter characteristic to the allelic subregion $\mathcal{R}_k$ and $\beta$ is a parameter characteristic to the entire allelic region $\mathcal{R}$. These parameters are estimated from given (incomplete) WGBS data via maximum-likelihood using Simulated Annealing (see Supplementary Methods, Section 5, for details).

An important objective of hap-ASM analysis is to identify haplotypes demonstrating statistically significant imbalances in DNA methylation stochasticity, which can be directly associated with differences in DNA sequence between homologous alleles. An equally important objective is to identify genetically informative haplotypes, defined as haplotypes in which the allele of origin conveys statistically significant information about the methylation state, suggesting that methylation stochasticity is closely associated with the allele of origin and therefore determined by the genetic variant. To this end, the CPEL method summarizes methylation stochasticity by using the mean methylation level (MML), which measures the average amount of methylation within an allele, the normalized methylation entropy (NME), which measures the amount of methylation stochasticity (pattern heterogeneity) within the allele, and the Jensen–Shannon distance (JSD), which quantifies differences between the two PDMs associated with the homologous alleles of a haplotype (see "Methods"). It then evaluates these statistical summaries by employing efficient algorithms (see Supplementary Methods, Sections 6–9, for details) and computes values of three test statistics,

$$T_{\mathrm{MML}} = |\mu(\mathbf{X}_1) - \mu(\mathbf{X}_2)|, \quad (4)$$

$$T_{\mathrm{NME}} = |h(\mathbf{X}_1) - h(\mathbf{X}_2)|, \quad (5)$$

and

$$T_{\mathrm{PDM}} = \frac{1}{N} \frac{D^2(p_1, p_2)}{h(\mathbf{X})}, \quad (6)$$

where $\mu(\mathbf{X}_1)$ and $\mu(\mathbf{X}_2)$ are the MMLs in the two homologous alleles of a given haplotype, $h(\mathbf{X}_1)$ and $h(\mathbf{X}_2)$ are the corresponding NMEs, $D(p_1, p_2)$ is the JSD between the associated PDMs $p_1(\mathbf{x})$ and $p_2(\mathbf{x})$, $h(\mathbf{X})$ is the NME of the methylation state $\mathbf{X}$ within the haplotype without knowing its allele of origin, and $N$ is the number of CpG sites. It finally uses these values to perform hap-ASM analysis by employing an one-sided empirical hypothesis testing procedure based on bootstrapping, which tests the null hypothesis that an observed test statistic value can be explained by the variability present in homozygous regions of the genome, against the alternative hypothesis that this value is due to a true allele-specific effect (see Supplementary Methods, Section 10, for details).

Besides to being proportional to the square JSD, the test statistic $T_{\mathrm{PDM}}$ in Eq. (6) computes the uncertainty coefficient (see "Methods"), which provides a measure of the amount of information that the allele of origin conveys about the random methylation state. Consequently, larger values of $T_{\mathrm{PDM}}$ indicate that the two PDMs corresponding to the homologous alleles of a haplotype are more distinct and that the random methylation state is more related to its allele of origin. Therefore, and in addition to detecting significant imbalances in MML and NME values within a haplotype by using the test statistics $T_{\mathrm{MML}}$ and $T_{\mathrm{NME}}$, the CPEL method identifies genetically informative haplotypes by determining significant PDM differences using the test statistic $T_{\mathrm{PDM}}$. Notably, $T_{\mathrm{MML}}$ and $T_{\mathrm{NME}}$ provide complementary insights into the statistical nature of methylation imbalances within a haplotype, since homologous alleles that exhibit similar levels of mean methylation may demonstrate significant differences in methylation entropies and vice versa, whereas $T_{\mathrm{PDM}}$ provides a more comprehensive measure of such imbalances in terms of differences between the PDMs associated with the two homologous alleles and its relationship with the uncertainty coefficient. We summarize the main steps (input, estimation, analysis, and output) of the CPEL method in Fig. 2.

**Simulation-based benchmarking**. To evaluate the performance of the CPEL method, we first used simulated data to assess its effectiveness for correctly estimating methylation probabilities in a haplotype when compared to the NPI and NPD methods.

We first considered a haplotype containing eight CpG sites and evaluated the performance of the three methods under five coverage regimes corresponding to 10, 20, 30, 40, and 50 fully observed methylation reads. We generated these reads by sampling a CPEL model with potential energy function given by Eq. (2), where $K = 1$, $\alpha_1 = 1$, and $\beta = 0$, thus producing reads in which the methylation states at individual CpG sites were mutually independent. Using these reads, we estimated the models associated with the NPI, NPD, and CPEL methods via maximum-likelihood and evaluated their goodness-of-fit by comparing the estimated PDM $\hat{p}(\mathbf{x})$ to the true PDM $p(\mathbf{x})$ using the JSD. By generating reads and by performing estimation 1000 times, we obtained the results depicted in Fig. 3a. Notably, all methods exhibited improved model estimation performance at increasing coverage. However, the NPD method produced poor estimates even at large coverage, whereas the CPEL method performed consistently well and noticeably better than the NPI method, even at low coverage.

We also studied estimation performance in terms of the number of CpG sites, by considering haplotypes containing 2, 4, 6, 8, and 10 CpG sites and by using 20 fully observed reads sampled from the previous CPEL model. According to the results depicted in Fig. 3b, all methods performed well in haplotypes containing two CpG sites. However, the performance of the two non-parametric methods, and especially that of the NPD method,

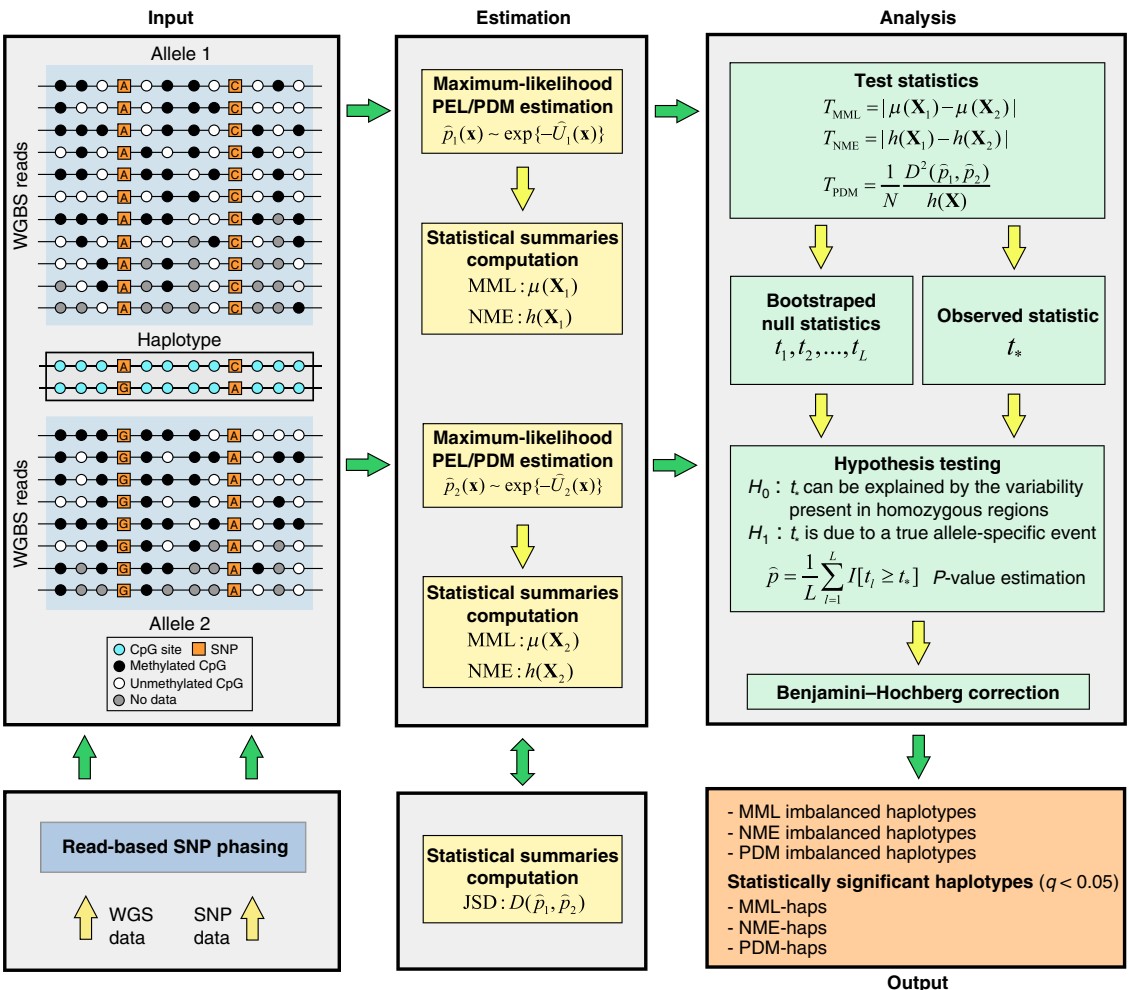

**Fig. 2 Flowchart illustration of the CPEL method.** The CPEL method first requires genome-wide identification of haplotypes, performed via read-based SNP phasing using WGS and SNP data, followed by mapping WGBS methylation reads to the homologous alleles of each haplotype according to their allele of origin. Using the WGBS data assigned to each allele, the CPEL method then computes a maximum-likelihood estimate $\widehat{U}(\mathbf{x})$ of the potential energy landscape (PEL) of the methylation state $\mathbf{x}$, given by Eq. (2), determines a PDM estimate $\widehat{p}(\mathbf{x}) \sim \exp\{-\widehat{U}(\mathbf{x})\}$, and summarizes methylation stochasticity in terms of the MML $\mu(\mathbf{X})$ and the NME $h(\mathbf{X})$ of the random methylation state $\mathbf{X}$, as well as in terms of the JSD $D(\widehat{p}_1, \widehat{p}_2)$ between the two estimated PDMs $\widehat{p}_1(\mathbf{x})$ and $\widehat{p}_2(\mathbf{x})$. Subsequently, the CPEL method performs hypothesis testing using the two test statistics $T_{\mathrm{MML}}$ and $T_{\mathrm{NME}}$ in Eqs. (4) and (5) in order to identify haplotypes demonstrating significant imbalances in MML and NME (MML-haps and NME-haps), as well as the test statistic $T_{\mathrm{PDM}}$ in Eq. (6), in order to identify haplotypes that exhibit significant differences between the two PDMs associated with their homologous paternal and maternal alleles (PDM-haps). To perform this step, the CPEL method uses an one-sided empirical bootstrap approach that estimates the $P$ value of a test by $\widehat{p} = (1/L)\sum_{l=1}^{L} I[t_l \geq t_*]$, where $t_1, t_2, ..., t_L$ are test statistic values appropriately drawn from the null distribution via bootstrapping, $t_*$ is the observed test statistic value, and $I[\,\cdot\,]$ is the Iverson bracket. Following a Benjamini–Hochberg step for multiple hypothesis testing correction, the CPEL method outputs three distinct lists of haplotypes, MML imbalanced haplotypes, NME imbalanced haplotypes, and PDM imbalanced haplotypes, together with their corresponding $Q$-values. Haplotypes associated with $Q$-values smaller than 0.05 are considered to be statistically significant.

greatly deteriorated as the number of CpG sites increased, while the CPEL method consistently demonstrated a good fit in all cases considered.

We subsequently performed simulations to investigate estimation performance when reads are subject to correlation. We considered again haplotypes containing 2, 4, 6, 8, and 10 CpG sites and used 20 fully observed reads sampled from a CPEL model with potential energy function given by Eq. (2), where $K = 1$, $\alpha_1 = 0$, and $\beta = 0.25$, 0.5, 0.75, 1. As expected, the results depicted in Fig. 3c and Supplementary Figs. 1a–c show that the NPI method performs poorly in the presence of correlations. Although the NPD method outperformed the NPI method when correlations were present, its estimation performance was not good, especially when compared to the CPEL method, which again behaved consistently well in the cases examined.

We further employed simulations to evaluate estimation performance when using partially observed reads, the most common scenario in WGBS. To that effect, we simulated 20 fully observed reads as before, randomly selected a subset of theses reads, and removed the methylation states of randomly selected CpG sites from each read in the subset. The results, depicted in Fig. 3d and Supplementary Fig. 1d, showed that the NPD method exhibits the worst behavior in this case, with its performance rapidly deteriorating as the number of CpG sites increased. Although the NPI method consistently outperformed the NPD method, it was also characterized by poor estimation performance, especially in the presence of correlations and in haplotypes with many CpG sites, while the CPEL method behaved again consistently well, clearly outperforming the other two methods.

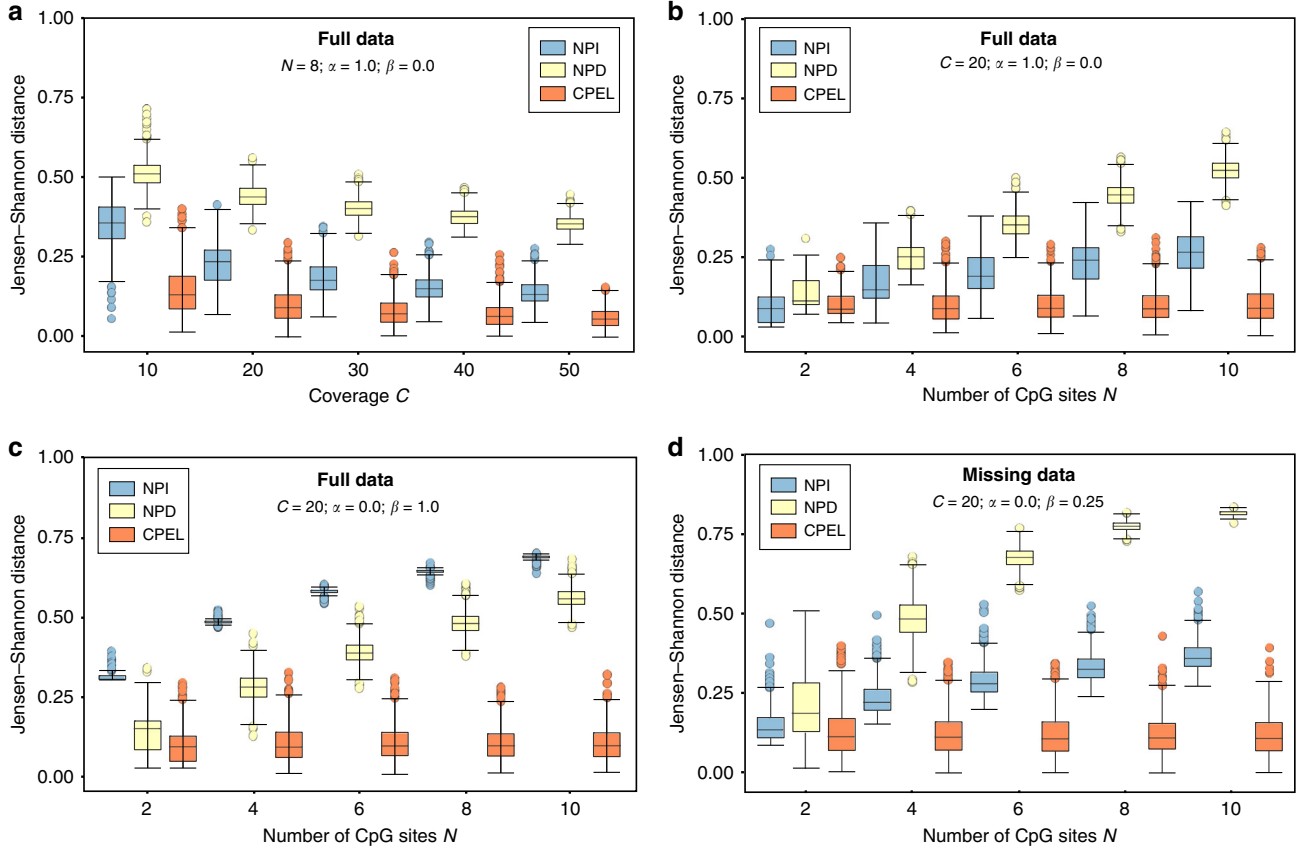

**Fig. 3 Simulated evaluation of model estimation performance.** Boxplots depicting distributions of Jensen–Shannon distance values when comparing estimated probability distributions of methylation (PDMs) to the true PDMs for a wide range of conditions using the NPI, NPD, and CPEL methods and simulated data. Estimation was independently performed 1000 times, each using fully observed or partially observed reads, as indicated. The results demonstrate a consistently superior performance of the CPEL method for correctly estimating methylation probabilities in a haplotype allele when compared to the NPI and NPD methods. **a** Fully observed non-correlated data and increased coverage. **b** Fully observed non-correlated data and increased number of CpG sites. **c** Fully observed correlated data and increased number of CpG sites. **d** Partially observed correlated data and increased number of CpG sites. Center line of box: median value; box bounds: 25th and 75th percentiles; lower whisker: larger of minimum value and 25th percentile minus 1.5 × interquartile range; upper whisker: smaller of maximum value and 75th percentile plus 1.5 × interquartile range.

In addition to the previous analysis, we examined whether the maximum-likelihood parameter estimator used by the CPEL method can reliably estimate the true values of the potential energy landscape parameters by performing estimation 1000 times for each case depicted in Fig. 3 and Supplementary Fig. 1. The results, summarized in Supplementary Fig. 2, demonstrated that the median of the parameter estimates almost always recovers the true values, whereas their variance decreases with increasing coverage or number of CpG sites, which is true even when incomplete data are available. However, and due to insufficient data, parameter estimation performed by the CPEL method may not always work as well when it comes to accurately estimating the true parameter values (see Supplementary Fig. 2, full or missing data with $C = 20$, $\alpha = 1.0$, $\beta = 0.0$, $N = 2$), an issue that will hamper parameter estimation regardless of the method used. Nonetheless, the results depicted in Fig. 3 and Supplementary Fig. 1 show that the PDMs estimated by the CPEL method remained close to the true distributions in our simulations even under this circumstance.

We also used simulations to investigate the quality of the hypothesis testing procedure employed by the CPEL method. Note that, by definition, if the null hypothesis of a statistical test is true, then the probability that a $P$ value is less than or equal to $a$ will be $a$ as well. This implies that, under the null hypothesis, a statistically sound hypothesis testing method must produce $P$ values that are uniformly distributed between 0 and 1. In this case, the method will provide proper control of the Type I error (false positives) under the null hypothesis, since the probability of obtaining a $P$ value that is below a significance level $a$ will precisely be $a$, implying a Type I error of $a\%$.

To investigate whether this is true for CPEL's hypothesis testing procedure, we performed simulations by considering the fact that, under the null hypothesis, the cumulative distribution function of a $T$ statistic associated with a given haplotype depends on the number $N$ of the CpG sites within the haplotype (see Supplementary Methods, Section 10, for details). To be consistent with the way the CPEL method estimates this function, we simulated a homozygous region containing $N$ CpG sites and assumed that the methylation states of its two homologous alleles are generated by a true CPEL model with $K_{\max}(N) + 1$ parameters, where $K_{\max}(N)$ is the maximum number of subregions observed within all haplotypes analyzed by the CPEL method in the real data. Under these conditions, we generated five methylation reads for each allele by sampling a true CPEL model with randomly determined parameter values, which produced methylation data within each homologous allele in a manner that was consistent with the coverage assumed by CPEL's hypothesis testing approach (see Supplementary Methods, Section 10, for details). Using these data, we finally computed an estimated CPEL model for each allele from which we

calculated, under the null hypothesis, a value $t$ for the $T$ statistic associated with the homozygous region.

By repeating the previous procedure, we obtained 1000 test statistic values, from which we derived an empirical estimate $\widehat{F}_0(t; N)$ of the cumulative distribution function of $T$ under the null hypothesis (see Supplementary Methods, Section 10, for details). We then computed a $P$ value for each value $t$ of $T$ by $\widehat{p} = 1 - \widehat{F}_0(t; N)$ and used these $P$ values to empirically estimate their cumulative distribution function. Our results, depicted in Supplementary Fig. 3 for $N = 2, 4, 6, 8, 10$, revealed that the empirically estimated cumulative distribution functions of the $P$ values for each of the three test statistics employed by the CPEL method were almost linear, implying that these $P$ values closely follow uniform distributions. This shows that the hypothesis testing approach employed by the CPEL method is indeed statistically sound producing a Type I error under the null hypothesis that is controlled by the test's significance level, as expected.

We finally used simulations to investigate the performance of CPEL's hypothesis testing procedure when applied on haplotypes that do not exhibit methylation imbalances. We did so by simulating 1000 haplotypes, each containing $N$ CpG sites. For each haplotype, we randomly chose the number of subregions to be between 1 and $K_{\max}(N)$ and the coverage within each homologous allele to be a random integer between 5 and 20. By following similar steps as before, we sampled a true CPEL model with $K_{\max}(N) + 1$ parameters to generate an appropriate number of methylation reads within each allele consistently with the associated coverage. We then computed an estimated CPEL model for each allele, calculated a value for the $T$ statistic under the null hypothesis, and estimated the corresponding $P$ value using the empirically estimated cumulative distribution function obtained from our previous simulations. For each $N = 2, 4, 6, 8, 10$, hypothesis testing found <5% of the haplotypes exhibiting statistically significant methylation imbalances ($P$ value $\leq 0.05$), demonstrating a Type I error that is smaller than the test's significance level and providing further evidence that the CPEL method performs hypothesis testing in a statistically sound manner.

**The CPEL method leads to superior modeling of real data.** We next evaluated the appropriateness of the NPI, NPD, and CPEL methods for modeling DNA methylation in real data. We did so by employing WGS, SNP, and WGBS data, previously used for ASM analysis by Onuchic et al.[12], corresponding to 10 different tissues from the same individual (see Supplementary Table 1). We found 715,155 haplotypes containing up to 121 SNPs and up to 20 CpG sites (see Supplementary Fig. 4). Using these haplotypes, we estimated the NPI, NPD, and CPEL models within each homologous allele, and identified all homologous alleles for which the CPEL model was more preferable than the other two models. We did so by quantifying the probability that a given model is more preferable for hap-ASM analysis than an alternative model using Akaike weights (see "Methods").

We first compared the CPEL and NPI models to discern which is best for the given data. Our results demonstrated that the CPEL model was more preferable overall than the NPI model in all samples, in the sense that its Akaike weight with respect to the NPI model was at least 0.5 in most haplotype alleles, when considering all haplotype alleles identified in the data (see Fig. 4a, yellow and gray). In addition, the CPEL model was distinctly superior to the NPI model when considering haplotype alleles containing at least four CpG sites (see Fig. 4b, yellow). By considering also the fact that the NPI model was not reliable at low coverage and in the presence of correlations in our simulations, we concluded that the applicability of the NPI method for hap-ASM analysis is limited.

We then compared the CPEL and NPD models to discern which was best for the real data. Since the NPD method cannot handle partial observations, we restricted our comparison to haplotypes with at least five full observations per haplotype allele, which resulted in analyzing only a small portion (about 16%) of all haplotypes identified in the data, as compared to 30% of the haplotypes analyzed by the CPEL method. Overall, our results demonstrated a clear preference for using the CPEL model over the NPD model within most haplotype alleles (see Fig. 4c, yellow and gray), with the CPEL model being distinctly superior to the NPD model when considering haplotype alleles containing at least four CpG sites (see Fig. 4d, yellow). When taking into account the fact that only 16% of the haplotypes could be analyzed by the NPD method, as well as its poor performance as the number of CpG sites increases, we reached the conclusion that this method is not appropriate for hap-ASM analysis.

We also investigated the quality of CPEL's hypothesis testing approach when applied to the real data. Similarly to our simulations, we found the empirically estimated cumulative distribution functions of the three test statistics employed by CPEL to be consistently linear under the null hypothesis (see Supplementary Fig. 5), thus leading to $P$ values that closely follow uniform distributions and demonstrating the ability of the CPEL method to control the Type I error under the null hypothesis (see Supplementary Fig. 6a). Notably, after applying the Benjamini–Hochberg procedure to control the false discovery rate, we identified no significant methylation imbalances (adjusted $P$ value $\leq 0.05$) in genomic regions that were not labeled to be haplotypes in the data (see Supplementary Fig. 6b). Finally, we considered the fact that large $P$ values associated with haplotypes that do not exhibit significant methylation imbalances in terms of the three test statistics must be samples drawn from a uniform distribution under the null hypothesis. In this case, we expect small observed $-\log_{10} P$ values to adhere to the diagonal of a Q–Q plot of observed vs. expected quantiles, which is shown to be true in the real data (see Supplementary Fig. 7), providing additional evidence for the statistical quality of CPEL's hypothesis testing procedure.

**Real data analysis.** To further evaluate the CPEL method, we analyzed results obtained by applying this approach on the real data. Distributions of MML and NME values associated with the two homologous alleles of identified haplotypes in each tissue from the same individual revealed that haplotype alleles are mostly partially methylated and subject to considerable methylation stochasticity, whereas only subtle differences were observed between the distributions associated with the two homologous alleles in a given tissue, as anticipated (see Supplementary Fig. 8). Interestingly, two tissues (intestine and pancreas) globally exhibited a noticeable loss of methylation level and gain in methylation entropy when compared to other tissues. Of the 715,155 haplotypes identified in the data across all tissues, only 90,524 (12.66%) were deemed to be statistically significant with 2320 of them exhibiting significant MML imbalances (MML-haps), 87,412 showing significant NME imbalances (NME-haps), and 1935 being identified as genetically informative demonstrating significant PDM differences within their homologous alleles (PDM-haps) that implies a significantly strong association between genetic variation and the methylation state (see Fig. 5a). As expected, distributions of MML-based test statistic ($T_{\text{MML}}$) values associated with significant and nonsignificant haplotypes in each tissue showed that MML-haps are subject to considerably larger MML imbalances than non-MML-haps, and the same was

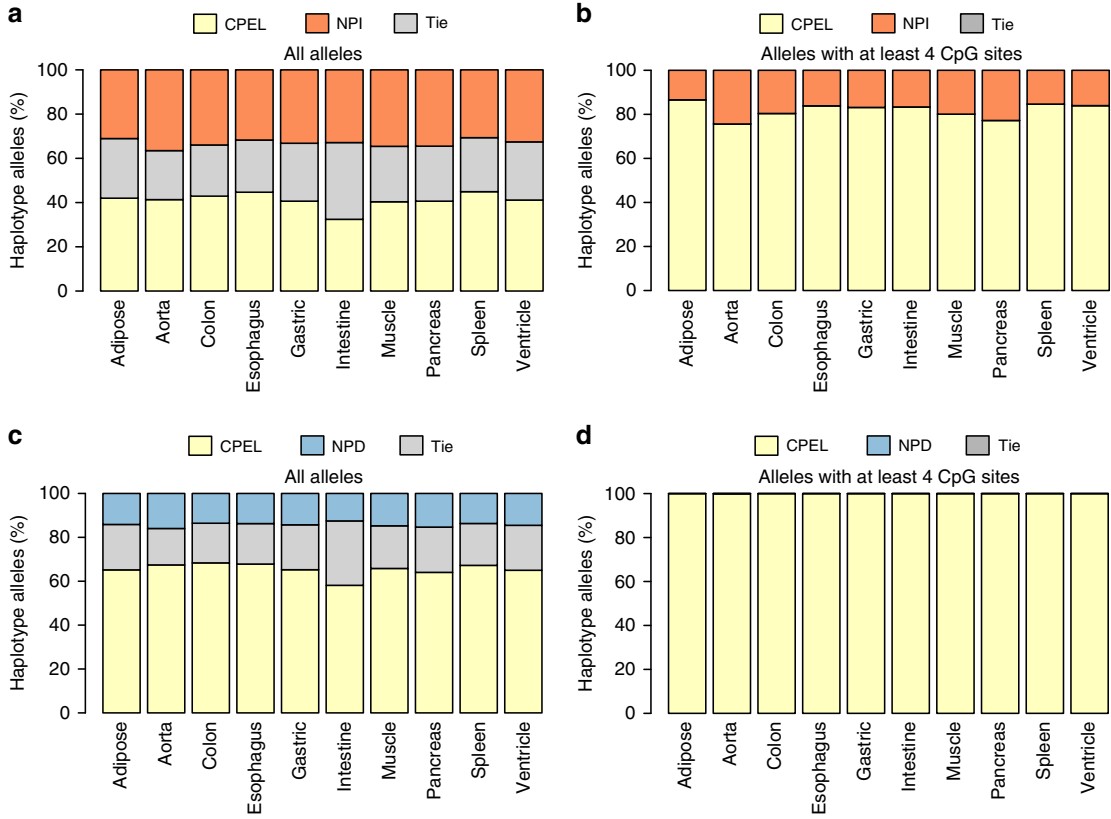

**Fig. 4 Evaluation of model fit to the real data.** Stacked histograms showing the percentage of haplotype alleles for which the CPEL model is more preferable (Akaike weight > 0.5), equally preferable (Akaike weight = 0.5), or less preferable (Akaike weight < 0.5) than the NPI or the NPD model. A total of 1,766,467 haplotype alleles have been considered. The results demonstrate the superiority of the CPEL model for fitting real WGBS data when compared to the NPI and NPD models, especially within haplotype alleles that contain at least four CpG sites. **a** Comparison between the CPEL and the NPI models based on all alleles. **b** Comparison between the CPEL and the NPI models based on alleles with at least four CpG sites. **c** Comparison between the CPEL and the NPD models based on all alleles. **d** Comparison between the CPEL and the NPD models based on alleles with at least four CpG sites.

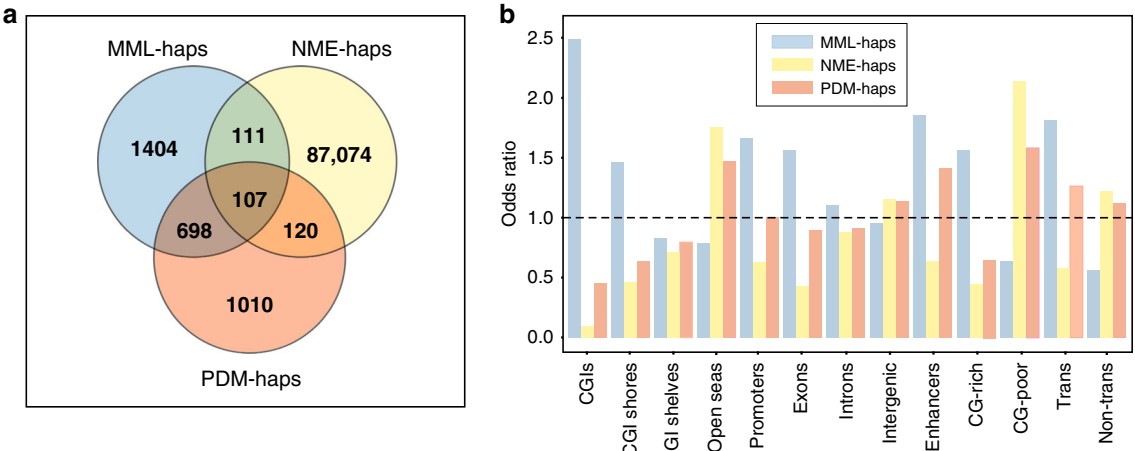

**Fig. 5 Results of real data ASM analysis using the CPEL method. a** Statistically significant haplotypes identified by the CPEL method across all tissues from the same individual in the real data and their specific attributes. **b** Odds ratio statistic values of statistically significant haplotypes overlapping genomic features and regions of interest across all tissues from the same individual. A total of 1,459,967 haplotypes have been considered. OR > 1 indicates enrichment, whereas OR < 1 indicates depletion. MML-haps: haplotypes exhibiting significant imbalances in mean methylation level; NME-haps: haplotypes exhibiting significant imbalances in normalized methylation entropy; PDM-haps: haplotypes exhibiting significant differences between the probability distributions of methylation within homologous alleles.

true for NME-haps and PDM-haps (see Supplementary Fig. 9). Notably, only a small fraction (1.55%) of statistically significant haplotypes exhibited imbalances in MML alone, whereas most significant haplotypes (96.19%) showed imbalances only in methylation entropy, demonstrating the potential importance of these haplotypes in ASM analysis. Moreover, 805 (41.60%) of the genetically informative haplotypes exhibited significant imbalances in MML, 227 (11.73%) showed significant imbalances in methylation entropy, and 107 (5.53%) demonstrated both (see Fig. 5a).

Using the previous results, we investigated tissue co-occurrence of statistically significant haplotypes. To that end, we defined the tissue co-occurrence of MML-haps to be the percentage of all haplotypes demonstrating significant MML imbalances in more than one tissue, and similarly for NME-haps and PDM-haps. We found that 15% of MML-haps in the real data co-occurred in more than one tissue, in agreement with known facts[5,6,13]. We also computed tissue co-occurrence for NME-haps, which has not been previously reported, and found that 34% of NME-haps co-occurred in more than one tissue, demonstrating that significant imbalances in methylation entropy co-occur across tissues more often than significant imbalances in MML. In part, this could be explained by our finding below that NME-haps tend to be located at CG-poor regions of the genome in which ASM stochasticity could be weakly regulated across tissues leading to consistently large non-tissue specific imbalances in methylation entropy within some haplotypes overlapping these regions. We also found that 10% of PDM-haps co-occurred in more than one tissue showing that genetically informative haplotypes (i.e., haplotypes that demonstrate significant PDM differences within their homologous alleles) co-occur less often than haplotypes exhibiting significant mean methylation imbalances or haplotypes demonstrating significant methylation entropy imbalances across tissues. Notably, the previous co-occurrences were found to be higher than what was expected by chance (all permutation test $P$ values < 0.001, see "Methods"), which is in line with recent results regarding co-occurrence of mean methylation imbalances across tissues[12]. Our results also revealed a high degree of tissue specificity, which we defined as the percentage of all statistically significant haplotypes of the same type (MML-haps, NME-haps, PDM-haps) that occur in only one tissue. We found 85% of MML-haps and 66% of NME-haps occurring in only one tissue, and the same was true for 90% of the PDM-haps, suggesting a possibly important role for PDM-haps in defining the phenotype.

We also evaluated enrichment of statistically significant haplotypes overlapping genomic features and regions of interest across all tissues using the odds ratio (OR) statistic and a Fisher's two-sided exact test (see "Methods"). Results obtained by the CPEL method (see Fig. 5b and Supplementary Tables 2 and 3) demonstrated a striking enrichment in MML-haps and depletion in NME-haps overlapping CpG islands (CGIs) contradicting previous suggestions of depletion of ASM events at CGIs[7,12], which we attributed to methodological limitations of those studies (see Supplementary Discussion). These findings suggest an association between CGIs and ASM events characterized by significant imbalances in MML but small differences in methylation entropy. Additional results (see Fig. 5b and Supplementary Tables 2 and 3) show a similar trend to that of CGIs for enhancers, gene promoter regions, exons, and CGI shores, albeit with smaller effect sizes. Notably, the enrichment result regarding enhancers is consistent with previous studies[12,14,15].

Overall, we found CG-rich regions of the genome, composed of CGIs and CGI shores, to be enriched in MML-haps but depleted in NME-haps and the same to be true for transcriptional regions that include promoters, exons, and enhancers (see Fig. 5b and Supplementary Tables 2 and 3). This points to a substantial

regulation of methylation within haplotypes overlapping CG-rich regions, more often than what is expected by chance, resulting in significantly imbalanced but highly ordered methylation states within their homologous alleles, thus producing small methylation entropy differences.

On the other hand, CG-poor regions, composed of CGI shelves and open seas, showed depletion in MML-haps but enrichment in NME-haps and the same was true for non-transcriptional regions (see Fig. 5b and Supplementary Tables 2 and 3). In part, we attributed these findings to a lesser need for precisely regulating methylation within haplotypes overlapping CG-poor and non-transcriptional regions of the genome, resulting in similar MMLs between the corresponding homologous alleles but significant imbalances in methylation entropy. We also found significant enrichment of PDM-haps overlapping CG-poor regions, open seas, and enhancers, and to a lesser extend transcriptional and intergenic regions (see Fig. 5b and Supplementary Table 4), demonstrating that genetically informative haplotypes overlap these regions more often than what is expected by chance. We hypothesized that identifying NME-haps and PDM-haps overlapping MML-hap-depleted CpG-poor and non-transcriptional regions of the genome can be important for understanding ASM, since ASM analysis using these haplotypes can potentially lead to better understanding the influence of genetic variation on ASM stochasticity in terms of high-order statistical summaries that go beyond the mean.

We further confirmed the validity of CPEL's output by checking whether the method identified statistically significant haplotypes overlapping the promoter regions of imprinted genes. This was motivated by the fact that control of certain imprinted genes may be facilitated by ASM at their promoters, since the imprinting control regions (ICRs) regulating imprint expression of some imprinted genes overlap their promoter regions. To that end, we tested for enrichment of statistically significant haplotypes overlapping the promoter regions of 107 known imprinted genes using the OR statistic and Fisher's two-sided exact test (see "Methods" and Supplementary Table 5). We found promoter regions of imprinted genes to be highly enriched in MML-haps and PDM-haps, indicating that haplotypes with significant imbalances in MML tend to occur, much more often than what is expected by chance, at the promoter regions of imprinted genes than at the promoter regions of non-imprinted genes, and the same was true for genetically informative haplotypes. However, we found no enrichment or depletion of haplotypes with significant imbalances in methylation entropy. This is consistent with the fact that the promoters of some imprinted genes may exhibit significant differential methylation between the two parental alleles, with one allele being associated with ordered methylation while its homologous allele exhibiting ordered demethylation, which results in significant mean methylation imbalances but nonsignificant entropy differences due to the low methylation entropies associated with two ordered methylation states.

To further investigate the relationship between imprinted genes and statistically significant haplotypes identified by the CPEL method, we found 53 genes with promoter regions that overlap haplotypes exhibiting significant imbalances in MML (MML-haps) in at least one tissue in the real data (see Supplementary Table 6). These genes included *GNAS*, *H19*, *MAGEL2*, *MESTIT1*, *NNAT*, *SNURF*, and *VTRNA2-1*, which are known to be imprinted. Although their promoter regions were found to overlap with 32 haplotypes across all tissues from the same individual, only 22 of them exhibited significant imbalances in MML (see Supplementary Table 7). Among these, 15 haplotypes (associated with *GNAS*, *H19*, *MAGEL2*, *MESTIT1*, *SNURF*, and *VTRNA2-1*) demonstrated significant imbalances

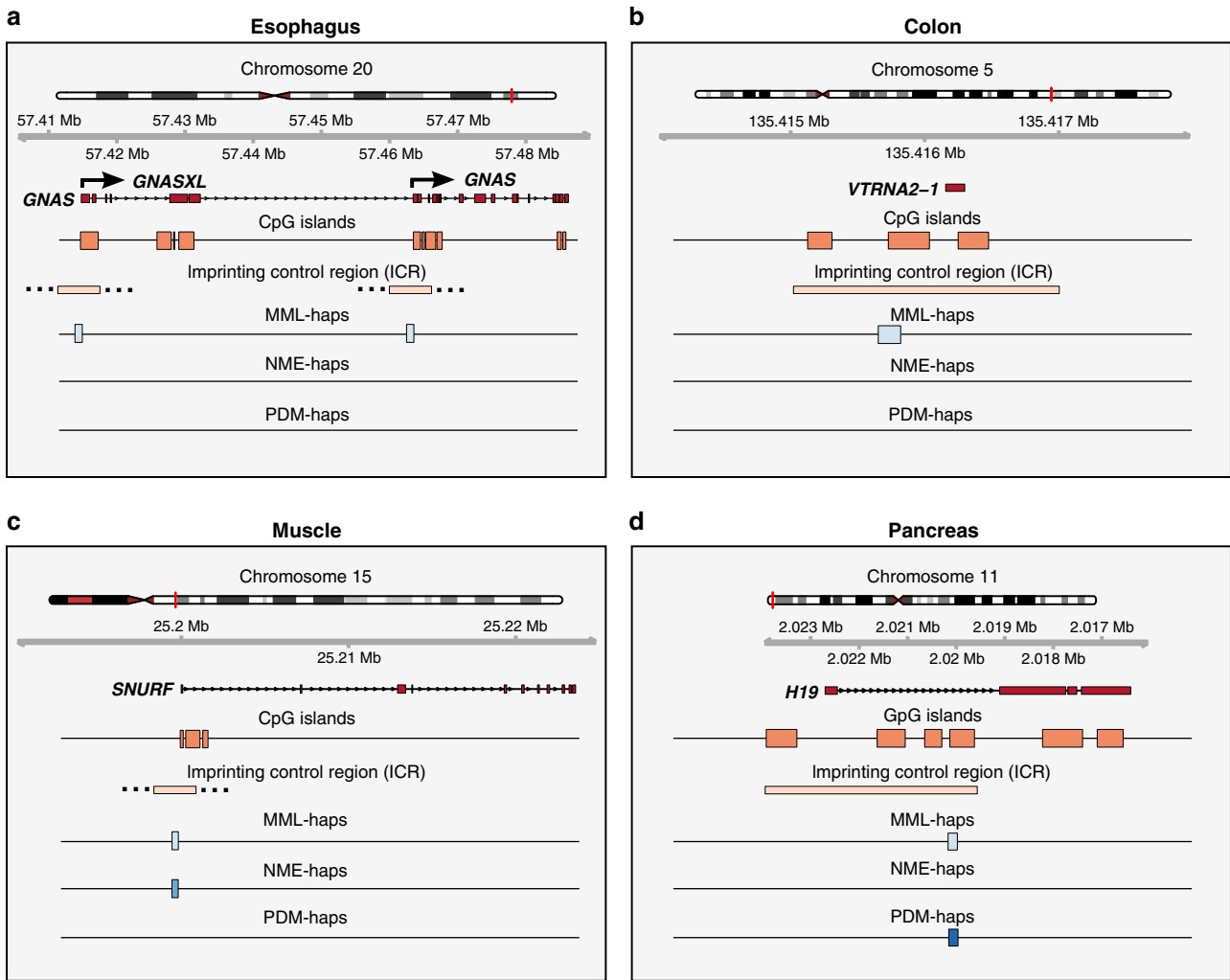

**Fig. 6 Illustrative examples of significant haplotypes and imprinting.** Haplotypes and their significance attributes overlapping the promoter regions of imprinted genes, identified by the CPEL method in different tissues of the same individual. **a** *GNAS* in esophagus, **b** *VTRNA2-1* in colon, **c** *SNURF* in muscle, **d** *H19* in pancreas. MML-haps: haplotypes exhibiting significant imbalances in mean methylation level; NME-haps: haplotypes exhibiting significant imbalances in normalized methylation entropy; PDM-haps: haplotypes exhibiting significant differences between the probability distributions of methylation within homologous alleles. The specific location of the imprinting control regions (ICRs) for *GNAS* and *SNURF* is not known.

only in MML, 2 haplotypes (associated with *MAGEL2* and *NNAT*) exhibited significant imbalances in MML and NME, and 5 haplotypes (associated with *H19*, *MAGEL2*, and *VTRNA2-1*) were found to be genetically informative.

To provide a more detailed picture of this behavior, we considered four illustrative examples (see Fig. 6). In the esophagus tissue, *GNAS*, an imprinted gene associated with the McCune-Albright syndrome and progressive osseus heteroplasia among other disorders[16], was associated with two haplotypes overlapping the promoter regions of two of its transcripts, *GNASXL* and *GNAS* (see Fig. 6a). Computed MML and NME values showed one allele in the first haplotype to be nearly methylated (MML = 0.87) and its homologous allele to be nearly unmethylated (MML = 0.16), whereas both alleles exhibited similar amounts of methylation stochasticity (NME = 0.54 and 0.46). This was also true for the second haplotype (MML = 0.86 and 0.08, NME = 0.45 and 0.40), resulting in both haplotypes exhibiting statistically significant imbalances in MML (MML-haps) but no significant differences in methylation entropy. Moreover, no significant differences between the corresponding PDMs were found in this case. Notably, the promoters of the

*GNASXL* and *GNAS* transcripts in mice are known to be embedded within two differentially methylated CpG island regions that carry characteristics of an imprinted control region[17].

Likewise, *VTRNA2-1*, an imprinted gene that appears to act as a tumor suppressor and linked to various types of cancer[18–20], was associated in the colon tissue with a haplotype that overlaps its promoter and ICRs (see Fig. 6b). Similarly to *GNAS*, this haplotype exhibited a statistically significant imbalance between its two homologous alleles only in MML (MML-hap), due to one allele being methylated (MML = 0.96) and its homologous allele being nearly unmethylated (MML = 0.17), with both alleles exhibiting similar amounts of methylation stochasticity (NME = 0.25 and 0.39). On the other hand, a haplotype was identified in the muscle tissue overlapping the promoter region of *SNURF* (see Fig. 6c), an imprinted gene associated with the Prader-Willi and Angelman syndromes[21]. This haplotype exhibited statistically significant imbalances in both MML (MML-hap) and methylation entropy (NME-hap) between its two homologous alleles (allele 1: MML = 0.97 and NME = 0.21; allele 2: MML = 0.25 and NME = 0.81) but no significant difference between the corresponding probability PDMs, indicating that the identified

haplotype is not genetically informative. Notably, the promoter region of *SNURF* is embedded within a 4.3-kb imprinting domain that is responsible for the Prader-Willi syndrome[22].

Finally, the imprinted gene *H19*, which has been associated with the Beckwith–Wiedemann and Silver–Russell syndromes[23], was linked in the pancreas tissue to a haplotype overlapping its promoter and ICRs (see Fig. 6d). Similarly to *GNAS* and *VTRNA2-1*, this haplotype exhibited a statistically significance imbalance in MML (MML-hap; MML = 0.97 and 0.17) but no significant difference in methylation entropy (NME = 0.16 and 0.45). However, it also demonstrated a statistically significant difference between the corresponding PDMs (PDM-hap), indicating that the identified haplotype is genetically informative in this case. Taken together, our real data results provide strong evidence that the CPEL method provides a powerful approach for hap-ASM analysis that can potentially produce valuable insights when assessing the impact of sequence-dependent allelic imbalances of the epigenome on cellular function.

## Discussion

Next-generation sequencing technology continues to improve read size and sequencing depth resulting in better experimental capabilities for measuring DNA methylation within large regions of the genome. This presents a great opportunity for performing ASM analysis within large haplotypes along the genome by jointly observing DNA methylation within homologous alleles at multiple CpG sites. However, the NPI and NPD methods, two state-of-the-art computational approaches for ASM analysis[7,12], have critical shortcomings that limit their applicability in this type of analysis (see Supplementary Discussion). Clearly, there is an immediate need for a new computational approach to ASM analysis that addresses the main drawbacks of existing methods. Toward this goal, we have developed the CPEL method, a powerful approach for hap-ASM analysis that leverages principles from statistical physics and information theory.

By using simulations, we have demonstrated the superiority of the CPEL method over the NPI and NPD methods. The NPI method is based on the biologically unrealistic assumption of statistical independence among the methylation states at individual CpG sites and, therefore, performs poorly in the presence of correlations. On the other hand, the performance of the NPD method deteriorates rapidly as the number of CpG sites within a haplotype increases. As a consequence, this method cannot perform ASM analysis reliably and reproducibly within large haplotypes containing multiple CpG sites. Notably, our results suggest that the ASM analysis approach of Onuchic et al.[12], which is based on the NPD method, can be subject to substantial statistical variability and, therefore, to questionable reproducibility. We attribute this issue to the seemingly poor estimation performance of the NPD method, especially in the presence of correlations and partially observed data, even at high coverage. In sharp contrast, the CPEL method appears to handle correlations and partially observed data well, even at low coverage and when haplotypes contain more than four CpG sites. Finally, the CPEL method can take into account heterozygous CpG sites that are created or removed by SNPs, which is not possible by current ASM analysis methods. This is done by estimating the PDM at all CpG sites within each homologous allele of a given haplotype using all available WGBS data and by computing PDMs for each allele over the homozygous CpG sites via marginalization, which are then utilized by the CPEL method to perform ASM analysis.

By employing WGBS data corresponding to 10 different tissues from the same individual and a model selection approach based on Akaike's information criterion (AIC), we have also demonstrated that the CPEL method is consistently more suitable for

ASM analysis than the NPI and NPD methods, especially when considering haplotypes with homologous alleles containing at least four CpG sites. This provides additional evidence that the CPEL method is more preferable than existing approaches to ASM analysis.

In addition to the previous limitations, the statistical analysis performed by the NPI and NPD methods focuses on detecting ASM events exhibiting significant differences in mean methylation, thus neglecting possibly critical ASM events associated with significant imbalances in methylation pattern variability between two homologous paternal alleles, as well as ASM events demonstrating a significant association between DNA sequence and the random methylation state. Although the CPEL method uses hypothesis testing to also detect haplotypes showing significant mean methylation imbalances, it can do so over larger genomic regions than existing methods. Moreover, it can detect haplotypes exhibiting statistically significant imbalances in methylation entropy (a statistical measure of methylation pattern variability), as well as identify genetically informative haplotypes characterized by significantly strong associations between genetic variation and the methylation state of the allele of origin. Note, however, that the clear superiority of the CPEL method over existing methods comes at a substantial computational cost (it took about 48 h using 20 CPUs to process one WGBS sample in the real data), which necessitates the use of parallel processing using a computer cluster. Given the cost of DNA sequencing as well as of data preprocessing and alignment, and considering the important statistical advantages that the CPEL method offers over existing approaches to ASM analysis, we believe that the added computational cost is a modest price to pay in most circumstances in order to obtain reliable and reproducible results.

The exploratory hap-ASM analysis performed here using the CPEL method demonstrates its potential for producing valuable biological insights, although more investigations are needed to further validate the biological results presented in this paper. For example, results obtained by analyzing the real data suggest that genetic differences are most often associated with significant methylation entropy imbalances in CG-poor and non-transcriptional genomic regions, while significant MML imbalances seem to generally occur at a smaller rate within CG-rich and transcriptional genomic regions. These two findings, which could shown be important for understanding the nature of allele-specific DNA methylation, cannot be obtained by existing approaches to ASM analysis, due to their previously discussed limitations.

Overall, the CPEL approach discussed in this paper improves previous work on ASM analysis and extends this work to large haplotypes containing many CpG sites. Most importantly, this method provides an essential set of innovative tools for comprehensively studying epigenetic stochasticity within homologous parental alleles and for quantifying and detecting imbalances in the statistical properties of such stochasticity along the genome. The keys to its success are threefold. First, the method comprehensively models the random methylation state at all CpG sites within a haplotype via a joint probability distribution that accounts for correlations among the methylation states of contiguous CpG sites. Second, this distribution is computed by estimating the values of a small number of parameters using all information available in the WGBS reads associated with the haplotype, a task that can be done reliably, even at low coverage. Third, the method incorporates an approach for statistical hypothesis testing that allows detection of significant differences in MML and methylation entropy, as well as identification of haplotypes for which the genetic variant carries significant information about the methylation state of neighboring CpG sites.

Providing researchers the capability to comprehensively perform hap-ASM analysis in a statistically rigorous manner constitutes a novelty in the ASM field. This opens the possibility of accurately studying the impact of genetic variation in *cis* on epigenetic stochasticity and ultimately facilitate the discovery of biological findings with important implications to the genetics of complex human diseases. Considering the advantages that the CPEL method enjoys over existing approaches, as well as the user-friendliness of the accompanying computer code, we believe that this method will be used widely for ASM analysis, leading to a better understanding of the circumstances in which ASM arises.

## Methods

**Data preprocessing and alignment**. We first performed quality control and adapter trimming of the raw WGS and WGBS data using Trim Galore (v0.5.0) [https://github.com/FelixKrueger/TrimGalore]. We then aligned the resulting WGS reads to the human reference assembly GRCh37 using Arioc[24] (v1.3.0) and removed PCR duplicates from the aligned reads using Picard[25] (v2.18). We performed a similar step on the raw WGBS reads using Arioc and a masked reference genome for alignment, as well as Bismark[26] (v0.20.0) for removing PCR duplicates. We created the masked reference genome by placing an undetermined base N at heterozygous SNP positions in the human reference assembly GRCh37 using SNPsplit[27] (v0.3.2). This ensured that the mapping score for either allele was not penalized due to a potential mismatch in the SNP position between a WGBS read and the reference genome. By utilizing available SNP information and by applying WhatsHap[28] (v0.17) on the aligned WGS reads, we performed read-based SNP phasing, which resulted in clustering SNPs in the same WGS reads into groups of the same genetic origin. Finally, by using SNPsplit, we classified aligned WGBS reads into three groups, according to their allele of origin or lack thereof.

**Identifying haplotypes**. An SNP cluster obtained by read-based SNP phasing defines a segment of DNA with a start point and an end point given by the genomic coordinates of its upstream-most and downstream-most SNPs, respectively. We call this region a haplotype and refer to each genetic variant of a haplotype as an allele. However, there might be nearby CpG sites upstream or downstream of a haplotype that should also be included in the comparison, given their proximity to the haplotype. To that end, the CPEL method symmetrically expands a haplotype on both sides (upstream and downstream) by including the same number of base pairs, which is taken to be the average size of the WGBS reads associated with the haplotype.

The CPEL method estimates *P* values from WGBS data, which turns out to be computationally expensive when haplotypes contain many CpG sites. To address this issue, a maximum acceptable number $N_{max}$ of CpG sites is specified and each haplotype that contains more than $N_{max}$ CpG sites is then divided into the minimum number of haplotype regions, with each region containing an equal number of CpG sites that is no more than $N_{max}$. In this case, the term haplotype refers to an individual haplotype containing at most $N_{max}$ CpG sites or to a haplotype region obtained by the previous approach. We implemented the CPEL method by setting $N_{max} = 20$, a choice that led to infrequent division of haplotypes in the real data and a reasonable usage of computational resources.

**Mean methylation level**. The average amount of methylation within an allele is quantified by the CPEL method using the MML[10]. For an allele associated with a homozygous haplotype containing $N$ CpG sites (i.e., one for which there is a perfect match between the CpG sites in the two homologous alleles), the MML is given by

$$\mu(\mathbf{X}) = \mathrm{E}\left[\frac{1}{N}\sum_{n=1}^{N} X_n\right], \tag{7}$$

where E[ · ] denotes expectation. The MML ranges between 0 and 1, taking its maximum value when all CpG sites within the allele are methylated and achieving its minimum value when all CpG sites are unmethylated. Efficient techniques for computing this quantity within homozygous and heterozygous haplotypes can be found in Supplementary Methods, Section 7.

**Normalized methylation entropy**. The amount of methylation stochasticity (pattern heterogeneity) within an allele is quantified by the CPEL method using the NME. For an allele associated with a homozygous haplotype containing $N$ CpG sites, the NME is given by

$$h(\mathbf{X}) = -\frac{1}{N}\sum_{\mathbf{x}} p(\mathbf{x})\log_2 p(\mathbf{x}), \tag{8}$$

with $\mathbf{X}$ being the allele's methylation state. The NME ranges between 0 and 1, taking its maximum value when all methylation states within the allele are equally likely (fully disordered methylation), and achieving its minimum value when only a single methylation state is observed (perfectly ordered methylation). Efficient

techniques for computing this quantity within homozygous and heterozygous haplotypes can be found in Supplementary Methods, Section 8.

**JSD and uncertainty coefficient**. Differences between two PDMs $p_1(\mathbf{x})$ and $p_2(\mathbf{x})$ are quantified by means of the JSD $D(p_1, p_2)$, where[10,11]

$$D(p_1, p_2) = \sqrt{\frac{1}{2}\left[\sum_{\mathbf{x}} p_1(\mathbf{x})\log_2 \frac{2p_1(\mathbf{x})}{p_1(\mathbf{x}) + p_2(\mathbf{x})} + \sum_{\mathbf{x}} p_2(\mathbf{x})\log_2 \frac{2p_2(\mathbf{x})}{p_1(\mathbf{x}) + p_2(\mathbf{x})}\right]}. \tag{9}$$

This quantity is a normalized metric since it takes values between 0 and 1, is symmetric, and satisfies the triangle inequality[29]. Moreover, it achieves its minimum value of 0 if and only if the two PDMs $p_1(\mathbf{x})$ and $p_2(\mathbf{x})$ are identical, whereas it takes its maximum value of 1 when the PDMs do not overlap with each other. In addition, the amount of information that the allele of origin $A \in \{1, 2\}$ conveys about the random methylation state $\mathbf{X}$ is measured by means of the uncertainty coefficient $Q(\mathbf{X}; A)$, given by[30]

$$Q(\mathbf{X}; A) = \frac{1}{N}\frac{I(\mathbf{X}; A)}{h(\mathbf{X})}, \tag{10}$$

where

$$I(\mathbf{X}; A) = -\sum_{\mathbf{x}}\sum_{a=1,2} \Pr[\mathbf{X} = \mathbf{x}, A = a]\log_2 \frac{\Pr[\mathbf{X} = \mathbf{x}, A = a]}{\Pr[\mathbf{X} = \mathbf{x}]\Pr[A = a]} \tag{11}$$

is the mutual information between the random methylation state $\mathbf{X}$ and the allele of origin $A$, $h(\mathbf{X})$ is the NME of $\mathbf{X}$ without knowing its allele of origin, and $N$ is the number of CpG sites. This quantity takes values between 0 and 1, with larger values indicating that the allele of origin conveys more information about the random methylation state.

Since we are considering diploid organisms, we can take the probability of finding one of the two homologous alleles of a given haplotype in a biological sample to be equal to the probability of finding the other allele; i.e., we can set $\Pr[A = 1] = \Pr[A = 2] = 1/2$. In this case, it can be shown[31] that $I(\mathbf{X}; A) = D^2(p_1, p_2)$, where $D(p_1, p_2)$ is the JSD between the PDMs $p_1(\mathbf{x}) = \Pr[\mathbf{X} = \mathbf{x}|A = 1]$ and $p_2(\mathbf{x}) = \Pr[\mathbf{X} = \mathbf{x}|A = 2]$ of the two homologous alleles. This implies that

$$Q(\mathbf{X}; A) = \frac{1}{N}\frac{D^2(p_1, p_2)}{h(\mathbf{X})} = T_{\mathrm{PDM}}, \tag{12}$$

which shows that, in addition to being proportional to the square JSD between the two PDMs $p_1(\mathbf{x})$ and $p_2(\mathbf{x})$ associated with the homologous alleles of a haplotype, the test statistic $T_{\mathrm{PDM}}$ used by the CPEL method provides a measure of association between the random methylation state and the allele of origin by means of the uncertainty coefficient. Efficient techniques for evaluating the uncertainty coefficient and, therefore, $T_{\mathrm{PDM}}$, within homozygous and heterozygous haplotypes are discussed in Supplementary Methods, Section 9.

**AIC and Akaike weights**. We evaluated the appropriateness of the NPI, NPD, and CPEL models for modeling real data using the small-sample version of AIC, given by[32]

$$\mathrm{AIC}(i, j) = -2\sum_{m=1}^{M_{ij}} \ln p_{\widehat{\boldsymbol{\theta}}_{ij}}(\mathbf{x}_m) + 2\eta_{ij} + \frac{2\eta_{ij}\left(\eta_{ij} + 1\right)}{M_{ij} - \eta_{ij} - 1}, \quad i = 1, 2, \ldots, j = 1, 2, \tag{13}$$

where $\eta_{ij}$ is the number of free parameters to be estimated for a given methylation model $p_{\boldsymbol{\theta}}(\mathbf{x})$ within the $j$th allele of the $i$th haplotype, $M_{ij}$ is the number of available observations, and $\widehat{\boldsymbol{\theta}}_{ij}$ are the estimated parameters. The AIC provides an estimate of the quality of a given ASM model by quantifying the relative amount of information lost when using this model, where a methylation model that is associated with a smaller information loss (i.e., a smaller AIC value) is considered to be better than a model that is associated with a higher information loss (i.e., a higher AIC value). Note that

$$\eta_{ij} = \begin{cases} N_{ij}, & \text{for the NPI model,} \\ 2^{N_{ij}} - 1, & \text{for the NPD model,} \\ K_{ij} + 1, & \text{for the CPEL model,} \end{cases} \tag{14}$$

where $N_{ij}$ is the number of CpG sites and $K_{ij}$ is the number of subregions in the $j$th homologous allele of the $i$th haplotype used by the CPEL model.

Given the AIC values $\mathrm{AIC}_1(i, j)$ and $\mathrm{AIC}_2(i, j)$ of two alternative ASM models associated with the $j$th allele of the $i$th haplotype, we computed the Akaike weight[32]

$$W(i, j) = \frac{\exp\{-\Delta_1(i, j)/2\}}{\exp\{-\Delta_1(i, j)/2\} + \exp\{-\Delta_2(i, j)/2\}}, \tag{15}$$

where $\Delta_1(i, j) = \mathrm{AIC}_1(i, j) - \min\{\mathrm{AIC}_1(i, j), \mathrm{AIC}_2(i, j)\}$ is the information loss experienced when using the first model rather than the best possible model (i.e., the model with the least AIC). By interpreting $W(i, j)$ as a probability, we concluded

that, when $W(i, j) > 0.5$, the first model was more preferable for hap-ASM analysis than the second model within the $j$th allele of the $i$th haplotype.

**Tissue co-occurrence permutation test**. We evaluated the significance of the tissue co-occurrence results using a permutation test, which tests against the null hypothesis that an observed co-occurrence value is no larger than what is expected by chance. This test counts the number $K$ of all statistically significant haplotypes of the same type (MML-hap, NME-hap, or PDM-hap) obtained by the CPEL method across all tissues. It subsequently performs 1000 permutations, each consisting of sequentially (up to $K$ times) and randomly labeling a haplotype along the genome and across all tissues to be statistically significant, and evaluates the corresponding tissue co-occurrence for each permutation. It finally computes a $P$ value as the proportion of tissue co-occurrences obtained from all permutations that are at least as large as the observed tissue co-occurrence. This $P$ value quantifies the probability of observing, under the permutation-based null model, a tissue co-occurrence at least as large as the one computed by the CPEL method. Therefore, a small $P$ value suggests that the observed tissue co-occurrence is less likely to be due to chance.

**Genomic features and regions**. Files and tracks bear genomic coordinates for hg19. We obtained CGIs from Wu et al.[33] and defined CGI shores as sequences flanking 2-kb on either side of CGIs, CGI shelves as sequences flanking 2-kb beyond the shores, and open seas as everything else. We divided the genome into CG-rich regions, composed of CGIs and CGI shores, and CG-poor regions composed of CGI shelves and open seas. We identified genes, transcription start sites (TSSs), exons, and introns using the R package `TxDb.Hsapiens.UCSC.hg19.knownGene` and defined the promoter region of a gene to be the 4-kb window centered at its TSS. We obtained enhancer annotations from a 25-state ChromHMM model[34], which produced 474,004 putative enhancer regions that we included in our analysis. We finally divided the genome into transcriptional regions, composed of promoters, exons, and enhancers, and defined non-transcriptional regions as everything else.

**Enrichment analysis**. We performed enrichment analysis of MML-haps overlapping a given genomic feature, and similarly for NME-haps and PDM-haps, by using the OR statistic and Fisher's two-sided exact test on a $2 \times 2$ contingency table of haplotype counts genome-wide and across all tissues with columns corresponding to the statistical significance of the haplotype (e.g., haplotype is MML-hap, haplotype is not MML-hap) and rows corresponding to haplotype overlap (haplotype overlaps genomic feature, haplotype does not overlap genomic feature). We implemented Fisher's two-sided exact test using the R function `fisher.test`.

We also used the previous method to perform enrichment analysis of MML-haps, NME-haps, and PDM-haps overlapping the promoter regions of imprinted genes. We employed 107 human genes from the geneimprint database [http://www.geneimprint.com] for which there is strong evidence that they are imprinted (i.e., whose status is labeled as imprinted in the database), and applied Fisher's two-sided exact test on a $2 \times 2$ contingency table of genome-wide haplotype counts across all tissues with columns corresponding to the statistical significance of the haplotype in the real data (e.g., haplotype is MML-hap, haplotype is not MML-hap) and rows corresponding to haplotypes overlapping the promoter region of an imprinted gene vs. that of a non-imprinted gene.

**Reporting summary**. Further information on research design is available in the Nature Research Reporting Summary linked to this article.

## Data availability

The raw WGS and WGBS data are available from NIH's Epigenomics Roadmap Initiative with number PRJNA34535 for patient ID STL003. The raw SNP data can be downloaded from Genboree with patient ID STL003 [https://fts.genboree.org/allelic-epigenome/json]. The homo sapiens (human) genome assembly GRCh37 (hg19) used as a reference genome can be downloaded from [https://hgdownload.soe.ucsc.edu/goldenPath/hg19/bigZips]. The gene imprinting data can be downloaded from [https://www.geneimprint.com/site/genes-by-species.Homo+sapiens.imprinted-All]. The enhancer data can be downloaded from [https://personal.broadinstitute.org/meuleman/reg2map/HoneyBadger2-intersect_release/DNase/p10/enh/25/state_calls.RData].

## Code availability

The method presented in this paper has been implemented in a `Julia` package called `CpelAsm`. The source code and associated instructions can be downloaded from [https://github.com/jordiabante/CpelAsm.jl].

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

## Acknowledgements

This work was supported by NSF Grant EFRI CEE 132452, NIH-NHGRI Grant RM1HG008529, and NIH-NIDDK Grant DP1DK119129. The funders had no role in study design, data collection and analysis, decision to publish, or preparation of the manuscript.

## Author contributions

J.A. and A.P.F. conceived the study. J.A. and J.G. developed the statistical and computational methods. J.A. implemented the methods and created the Julia package. Y.F. procured outside data, performed quality control, preprocessing, whole-genome and whole-genome bisulfite alignment, as well as SNP phasing and allele-specific read assignment. J.A. and J.G. interpreted the results and wrote the manuscript with the assistance of Y.F. and A.P.F.

## Competing interests

The authors declare no competing interests.
