## [Peer Review File · Nature Communications]

Reviewers' comments:

Reviewer #1 (Remarks to the Author):

The manuscript by Abante et al. entitled "Detection of haplotype-dependent allele-specific DNA methylation in WGBS data" explores a very interesting question. Is it possible to infer haplotypes in DNA methylation using a combination of WGS and WGBS? The authors developed a Julia package, unfortunately after downloading the zip file it is corrupted (the file downloaded is 114KB is this correct for the code, examples, etc.?, it seems tiny), so I could not evaluate the contents of the code or the implementation of the examples. A similar problem occurred when trying to download the data files provided as a zip file. I would recommend some detailed instructions for the reviewers about the contents, installation and dependencies to be able to provide any feedback as the github is unavailable. The authors adapt a one-dimensional Ising model that provides a framework for phase transitions into the CPEL (correlated potential energy landscape). If successful, this is an important model for future discoveries in the field. The hypothesis of the authors is that DNA methylation could present in some contexts allele-specific methylation, this is based on Kerker et al observations of DNA methylation in specific C to T polymorphisms in some targeted areas or as reported by several authors (e.g. Schalkwyk et al.) due to parental imprinting and cis-genetic variation imbalance. Combining WGS to obtain the polymorphisms with WGBS to obtain the methylation state it is possible to determine the allele specific methylation. The authors provide a valuable context of the statistical assumptions of previous methods exploring this problem too.

Comments:

Figure 1: The graphical explanation of the three methods is very interesting, but the caption is not explaining what is happening in the figure. The epialleles are clearly the two strands, but the WGBS are the number of reads?. I would recommend that you simplify the caption and explain the colors and contents of the figure for the reader.

Page 4, paragraph 2: "Insufficient coverage can lead to unreasonably high statistical variability when NPD is used for hap-ASM analysis, resulting in very low statistical power and a high rate of false positives." Do you have any quantification of the magnitude of the error? could you provide an example or reference for this claim? This is expanded in the comparison later in the text after page 11.

Figure 2: What is TPDM? The test for paternal vs maternal alleles? Again the summary is very interesting, but I would double check to help the reader to follow all the process as the notation and multiple acronyms were difficult to follow through the text. The notation initially seem clear, but the inputs X , x , A should be transparent for the reader, right now it is difficult to follow the inputs in the figure. This is clearer in the text in page 9, but it is difficult to follow in the figure which is presented earlier in the manuscript.

Page 15 Real data analysis, Figure 5: Imbalances in methylation levels in the haplotypes was a very small fraction (~1%), most were due to methylation entropy (~96%) stochasticity of the pattern. One pattern that called my attention was that co-occurrence of PDM imbalances only occurred in 10% of the tissues. How can you ascertain that this is a real allele specific imbalance in contrast with a DNA methylation change between different cells/tissues? This is reiterated in page 16 mentioning the 90% tissue specificity. If this is real this is huge, but I do not see how these results are supported by other experimental gold-standard comparisons. Do you have the means to perform or analyze data with some gold-standard as a comparison. Right now this claim is only supported by the model, but not for other experiments, could you expand on how to sustain this claim or moderate the message here?

Page 16: "Notably, we found the previous co-occurrences to be higher than what is expected by chance (all permutation ' test p-values = 0, see Methods)," Did you mean by chance?

Page 16: Reading your supplementary material you found several known imprinted genes, plus several new genes. I would advise that some of this material should be moved to the main text to connect the dots between your method and the biology, even moving some of the figure 5 earlier could help.

Supplementary table 2 and 3: Could you check the p-values and extract the actual p-value instead of the table limit in R or similar. Why do you have zeroes in the supplementary table 3. Please double check those.

Pages 19, supplementary table 5: You tested for enrichment the human genes known and validated as imprinted regions on your supplementary table 5 (n=107 genes). Of those how many were detected by the algorithm (only 7?)? What about the other less know genes from geneimprint (n= 150)? How many of those did you find in your data? how many were skipped? Could you expand why some of the known imprinted genes were or were not detected by the algorithm (coverage, sequencing problems, others)? Please also correct the p-value from the supplementary table 5.

Page 20: I am curious about how the algorithm worked on the X chromosome in females? Or did you excluded this area from the analyses? I do not see this explicitly in your methods. Could you expand what will you expect on those areas?

Reviewer #3 (Remarks to the Author):

Allele-specific methylation (ASM) analysis is important for studying association between perturbed methylation and human diseases, but far from trivial. In this work, Abante et al propose a new approach that identifies haplotype-dependent ASM events. Their 1-dim Ising model, CPEL, enables the joint profiling of methylation state across multiple CpG sites. Since the methylation probability is defined at the read level, CPEL provides merits over existing methods. For example, the model is robust in either cases where the methylation of neighboring sites are dependent or independent. Using the concept of entropy as another difference measure for allelic methylation state is unique. Using homozygous sites to form a null distribution is also novel. The manuscript is clear and evidence provided is convincing. The manuscript can be accepted as is in my opinion, but it may still improve its scientific value if the authors address the following points:

Major points:

1. Are the SNPs included when profiling methylation state? If so, how does bismark-based strategy distinguish methylated C from unmethylated T allele? If not, what would you do differently in CPEL to include them?
2. How would the proposed method scale when haplotype blocks are very large as in F1 hybrid mouse? It would be very interesting to see how CPEL would perform on a dataset by Gigante et al 2019 NAR (ERP109201).
3. Please show the distribution of MML, NME, and PDM.
4. Please show if CPEL recovered parameter values that are used in the simulation.
5. Is it feasible to run simulated annealing many times to see the stability of parameter

estimation?

Minor points:

1. Clarify what "this important issue" is in Line 15 Page 10.
2. Remove NPI, NPD, and CPEL from Figure 1 legend.
3. Define what A is in Equation (5).

Reviewer #4 (Remarks to the Author):

The paper described a new method, CPEL, for detecting allele-specific methylation events on haplotypes. CPEL relies on a simplified version of the authors' previous Ising model to model methylation events on genomic segments. CPEL examines one haplotype at a time and uses bootstrap to test the difference between the two haplotype alleles in terms of (a) mean methylation level (MML), (b) normalized methylation entropy (NME) which quantifies methylation variability across CpG sites on the haplotype, and (c) probability distributions of methylation (PDMs). CPEL is applied, along with two existing methods NPI and NPD, to examine allele specific methylation events in a real data set. Overall, the paper is well written, and the adaption of their previous method for allelic specific methylation analysis is simple but of interest. However, it is unclear at the moment whether CPEL is a valid statistical approach that can provide well calibrated type I error control. In addition, the detected MML-hap enrichment in the CpG islands in the real data applications is a bit concerning as it contradicts all existing literature. My main comments are listed below:

1. CPEL uses a relatively complicated bootstrap process to obtain p-values for statistical test. It is unclear, however, whether such test is calibrated in terms of type I error control. Demonstrating the CPEL can provide well controlled type I error is particularly important here, given that the identified MML-haps by CPEL are somehow highly enriched in CpG islands (CGIs), which contradicts almost all previous literature. Therefore, it would be important to show that CPEL controls type I error well both in simulations and in real data applications. Specifically:
 - 1.1 It would be useful to simulate the null data under the Ising model, and then perform a similar bootstrap procedure as used in the real data, to examine whether the type I error is well controlled in CEPL (as well as in NPI and NPD). It would be important to look at type I error control at the genome-wide significance level used in the real data applications.
 - 1.2 It would be important to provide the $-\log_{10}$ p-value distribution in the real data through qq-plots, for each of the three tests (MML, NME, PDM) and in each tissue. Because the majority of the haplotypes do not show significance in the real data and are effectively null, one would expect the small $-\log_{10}$ p-values to adhere on the diagonal line on the qq-plot. In addition, it would be important to report the genomic control factors for each test and in each tissue. A genomic control factor that is close to 1 would indicate reasonably good type I error control in the real data analysis.
 - 1.3 Some part of the bootstrap procedure is not realistic and can be improved. For example, right now the reads are placed into two complementary groups that contain the same number of reads. The constraint that the two groups contain the same number of reads seems rather unrealistic. A better choice would be to randomly assign reads onto these two group without such equal read constraint. This can be easily achieved by assigning each read onto two haplotypes through an over-dispersed binomial distribution (beta-binomial) with mean of 0.5. This more realistic strategy could help improve type I error control and reduce potential false discoveries.
2. A key potential bias for allelic specific methylation analysis is due to read mapping. Specifically, when a CpG site is in a neighborhood of a SNP, it is much harder to map a methylated read to the

alternative allele than an unmethylated read. Consequently, methylation level for the given CpG site maybe artificially higher on the haplotype with the reference SNP, leading to methylation bias towards the reference allele. This could cause potentially false signals in the NME analysis, and, likely to a lesser extent, in the MML analysis. If you plot methylation levels of the CpG sites on the alternative SNP allele vs the reference SNP allele, would you see a bias? If there is a mapping bias, does the mapping bias occur preferentially on CGIs (or other regions with high CpG density), so that it creates a false enrichment of MML-haps on CGIs? The mapping bias can be mitigated, though not completely eliminated, by relaxing the mismatches during the reads mapping step. The mapping bias can also be mitigated by excluding CpG sites with an obvious difference in methylation levels between reference and alternate alleles. I am a bit concerned whether such mapping bias may introduce false signals in the MML and NME analysis.

3. It would be important to apply NPI and NPD to all enrichment analysis in the real data and compare their results with CPEL. It is equally important to examine the ability of NPI and NPD in terms of identifying significant regions overlapping the promoter regions of imprinted genes, across different tissues. Comparing CPEL with NPI and NPD on imprinted genes would provide a much stronger evidence for the statistical power difference between CPEL and the other two methods in the real data applications.

4. CPEL is effectively a simplified version of their previous method (introduced in ref #10). The simplifications include (a) removing the dependence of alpha on CpG density; (b) remove the dependence of beta on distance between pairs of CpG sites. These simplifications seem counter-intuitive, given that both features (a) and (b) are rather desirable for real data applications. Therefore, it would be important to compare CPEL with their previous method in all real data applications to demonstrate the benefits of removing features (a) and (b).

5. A critical parameter in the CPEL method is the subregion size, denoted by G. G was set to be 500bp. How stable are the real data results with respect to the choice of G? Would these real data results change if you use a smaller G or a larger G?

6. Is there a way to quantify whether these detected allelic specific methylation events are due to cis genetic differences? For example, do you tend to detect haplotypes with a high number of SNPs, a high density of SNPs, or a long region length?

7. In the real data applications, do detected regions of either MML, NME, or PDM correlated with TAD boundaries?

8. It would be useful to report the computation time of CPEL in real data applications. In addition, the bootstrap procedure in CPEL only keeps bootstrapped sites where CPEL estimation is successful. What's the success rates of this step in the bootstrap in real data applications?

RESPONSE TO REVIEWERS

We would like to thank the reviewers for the interest they showed in reading our paper and for providing their comments. This is a very time-consuming process and we greatly appreciate their thoughtful reviews. We provide our responses below.

REVIEWER #1

1. ***The authors developed a Julia package, unfortunately after downloading the zip file it is corrupted (the file downloaded is 114KB is this correct for the code, examples, etc.?, it seems tiny), so I could not evaluate the contents of the code or the implementation of the examples. A similar problem occurred when trying to download the data files provided as a zip file.***

We are surprised that the reviewer was unable to open the zip file, since we did not find anything wrong with it. The size of the zip file is small (the new version is 216 KB). This file contains the CpelAsm software as well as a README text file that provides info about installation and a small synthetic example based on simulated data. This will allow the reviewer to examine the actual source code, install the software on their computer (although access through GitHub will do the installation automatically using a single command), test the code on the example, and see the results. We do not provide an example using real data, since this requires using a computer cluster (see our response to #11 of Reviewer 4). We considered providing a small part of the real data (one chromosome, for example) instead of a synthetic example, but this idea did not work since correct implementation of CpelAsm requires estimation of the null statistics, which is done genome-wide. To make sure that the reviewer receives the zip file, we have also e-mailed a copy to the Senior Editor.

2. ***I would recommend some detailed instructions for the reviewers about the contents, installation and dependencies to be able to provide any feedback as the github is unavailable.***

Detailed instructions about the installation of the software and running the provided example can be found in the README.txt file submitted as part of 'CpelAsm-code.zip' file.

3. ***Figure 1: The graphical explanation of the three methods is very interesting, but the caption is not explaining what is happening in the figure. The epialleles are clearly the two strands, but the WGBS are the number of reads? I would recommend that you simplify the caption and explain the colors and contents of the figure for the reader.***

We simplified the caption and better explained the colors and content of the figure. We hope that now the caption is more clear and easier to understand. We also slightly modified the figure to accommodate the new caption.

4. ***Page 4, paragraph 2: "Insufficient coverage can lead to unreasonably high statistical variability when NPD is used for hap-ASM analysis, resulting in very low statistical power and a high rate of false positives." Do you have any quantification of the magnitude of the error? Could you provide an example or reference for this claim? This is expanded in the comparison later in the text after page 11.***

An extensive discussion on the magnitude of the statistical variability occurring when the NPD method is used to estimate epiallelic probabilities can be found in Section 3 of the Supplementary Note in [1]. According to the thorough analysis performed there, the NPD method will not result in accurate estimates of the epiallelic probabilities when low coverage data is used (the case of interest), especially in areas of the genome that exhibit high methylation stochasticity, resulting in substantial uncertainty in the estimates. In addition, this problem will become worse as the number of CpG sites included in the analysis increases, which follows from the fact that the number of epiallelic patterns grows geometrically with the number of CpG sites. This problem will seriously affect downstream statistical analysis, especially when this analysis is based on high order statistical quantities, such as entropy. Note that if such a quantity is to be used as a test statistic to perform hypothesis testing, larger variability in the test statistic will hinder the power of the test significantly, since larger values of the test statistic will be required in order to reject the null hypothesis. This can be critical when the resulting statistical variability overwhelms the underlying biological variability, hiding any real biological effect. In addition, one might be tempted to increase the significance level of the test in order to detect differences, given the large variability in the test statistic, resulting in a larger number of false positives.

To briefly discuss these issues, we added on pages 4-5 of the revised Main Text the following:

“Insufficient coverage can lead to large uncertainty and low accuracy when estimating epiallelic probabilities using the NPD method, especially in areas of the genome that exhibit high methylation stochasticity (see Jenkinson et al¹⁰, Supplementary Note, Section 3), which can seriously affect downstream statistical analysis. This problem is exacerbated when epialleles that contain more than 4 CpG sites are used in the analysis, due to the geometric growth of the number of epiallelic patterns associated with an increasing number of CpG sites. As a consequence, the NPD method is not appropriate for hap-ASM analysis, which often requires estimation of joint methylation probabilities within genomic regions that contain more than 4 CpG sites.”

[1] G. Jenkinson, E. Pujadas, J. Goutsias, and A. P. Feinberg, “Potential energy landscapes identify the information-theoretic nature of the epigenome.” *Nature Genetics* 49(5), 719-729 (2017).

- 5. *Figure 2: What is TPDM? The test for paternal vs maternal alleles? Again the summary is very interesting, but I would double check to help the reader to follow all the process as the notation and multiple acronyms were difficult to follow through the text. The notation initially seem clear, but the inputs X, x, A should be transparent for the reader, right now it is difficult to follow the inputs in the figure. This is clearer in the text in page 9, but it is difficult to follow in the figure which is presented earlier in the manuscript.***

We now reference Figure 2 on page 11 of the revised Main Text. At this point, the reader should be familiar with the notation and the quantities used in this figure. We have modified the caption to make it more clear and complete. The caption now provides a summary of the four steps in the CPEL method, defines the acronyms, and refers to appropriate equations in the text. Note that, throughout the manuscript, we use a standard convention in probability theory: we denote a random variable by an upper case letter, such as X, and its realization (observed value) using a lower case letter, such as x. We have added a note to that effect in the Main Text; see page 7 of the

revised Main text. We also made clear what A represents in Eq. (5) by specifying what it is on page 10 of the revised Main Text.

- 6. Page 15 Real data analysis, Figure 5: Imbalances in methylation levels in the haplotypes was a very small fraction (~1%), most were due to methylation entropy (~96%) stochasticity of the pattern. One pattern that called my attention was that co-occurrence of PDM imbalances only occurred in 10% of the tissues. How can you ascertain that this is a real allele specific imbalance in contrast with a DNA methylation change between different cells/tissues? This is reiterated in page 16 mentioning the 90% tissue specificity. If this is real this is huge, but I do not see how these results are supported by other experimental gold-standard comparisons. Do you have the means to perform or analyze data with some gold-standard as a comparison. Right now this claim is only supported by the model, but not for other experiments, could you expand on how to sustain this claim or moderate the message here?**

The concern raised by the reviewer stems from the following statements: “co-occurrence of PDM imbalances only occurred in 10% of the tissues” and that “This is reiterated in page 16 mentioning the 90% tissue specificity.” These statements misinterpret our co-occurrence and specificity results, which are not directly related to Figure 5. Our statement in our previous version of our paper stated that “We also found a 10% tissue co-occurrence for PDM-haps”. Using the definition of tissue co-occurrence we provided in the Main Text (the percentage of all MML-haps demonstrating significant MML imbalances in more than one tissue, and similarly for NME-haps and PDM-haps), this result should be interpreted as “10% of PDM-haps co-occurred in more than one tissue”. To avoid confusion, we replaced our previous statement with “We also found that 10% of PDM-haps co-occurred in more than one tissue,” and did the same for the co-occurrence statements regarding MML-haps and NME-haps. See page 20 of the revised version of the Main Text.

Similarly, using the definition of tissue specificity we provided in the Main Text [the percentage of all statistically significant haplotypes of the same type (MML-haps, NME-haps, PDM-haps) that occur in only one tissue], the 90% tissue specificity for PDM-haps should be interpreted as “90% of PDM-haps occur in only one tissue.” To avoid confusion, we replaced our previous statement with “We found 85% of MML-haps and 66% of NME-haps occurring in only one tissue, and the same was true for 90% of the PDM-haps, suggesting a possibly important role for PDM-haps in defining the phenotype.” See pages 20-21 of the revised version of the Main Text.

- 7. Page 16: "Notably, we found the previous co-occurrences to be higher than what is expected by change (all permutation' test p-values = 0, see Methods)," Did you mean by chance?**

Yes, “chance” not “change.” We fixed this typo.

- 8. Page 16: Reading your supplementary material you found several known imprinted genes, plus several new genes. I would advise that some of this material should be moved to the main text to connect the dots between your method and the biology, even moving some of the figure 5 earlier could help.**

We moved this material to the Main Text, as suggested; see pages 23-24. We did so by generating a new figure (Fig. 6), which now includes the imprinting results depicted in the old Fig. 5c, as well as

the ones included in the old Supplementary Figs. 3-5. The new Fig. 5 is now limited to the information depicted in the old Figs. 5a,b.

- 9. Supplementary table 2 and 3: Could you check the p-values and extract the actual p-value instead of the table limit in R or similar. Why do you have zeroes in the supplementary table 3. Please double check those.**

We have revised the *P*-values in the new version of our paper following suggestions published in a recent paper [1]. We now report the actual *P*-values only when they are at least equal to 0.001 and all other *P*-values as < 0.001 . According to the previous paper, "There is little practical difference among very small *P*-values when the assumptions used to compute *P*-values are not known with enough certainty to justify such precision, and most methods for computing *P*-values are not numerically accurate below a certain point." In particular, we follow the convention of the *New England Journal of Medicine* (NEJM), which states: "In general, *P* values larger than 0.01 should be reported to two decimal places, those between 0.01 and 0.001 to three decimal places; *P* values smaller than 0.001 should be reported as $P < 0.001$."

[1] S. Greenland, S. J. Senn, K. J. Rothman, *et al.* "Statistical tests, *P*-values, confidence intervals, and power: a guide to misinterpretations." *The American Statistician* 31(4), 337-350 (2016).

- 10. Pages 19, supplementary table 5: You tested for enrichment the human genes known and validated as imprinted regions on your supplementary table 5 (n=107 genes). Of those how many were detected by the algorithm (only 7)? What about the other less known genes from geneimprint (n= 150)? How many of those did you find in your data? How many were skipped? Could you expand why some of the known imprinted genes were or were not detected by the algorithm (coverage, sequencing problems, others)? Please also correct the p-value from the supplementary table 5.**

We chose to use only the 107 human genes in www.geneimprint.com whose status have been classified as being 'imprinted' for which there is strong evidence that are indeed imprinted. The rest of the genes reported in that database are either based on predictions ('predicted'), subject to either conflicting or provisional data ('conflicting data', 'provisional data'), not imprinted at all ('not imprinted'), or unknown ('unknown'). We did so to reduce the amount of noise introduced in our enrichment analysis. This allowed us to show a strong association between MML-haps and PDM-haps that might have been otherwise diluted if we had included genes of dubious imprinting status to the statistical analysis. We tried to make this clear in our Methods section by writing

"We employed 107 human genes from the geneimprint database (www.geneimprint.com) for which there is strong evidence that they are imprinted (i.e., whose status is labeled as 'imprinted' in the database)".

See page 36 of the revised Main Text.

As the reviewer points out, our algorithm detected 7/107 genes from the geneimprint database. There are several reasons why this is the case. First, ASM analysis using our method can only be done when there is available data in the corresponding region. Second, gene imprinting is known to be tissue specific. As such, it is possible that some genes from the geneimprint database are not actually imprinted in the tissues considered in our work and, therefore, we do not expect to detect

allelic imbalances when this is this case. Finally, although ASM at promoter regions is a frequent trait of imprinted genes, this trait is not a necessary condition for an imprinted gene [1]. We could investigate the specific case for each gene in the database, but the results will not add any useful information to our analysis. Nevertheless, our enrichment analysis shows that there is strong enrichment of MML-haps and PDM-haps at promoter regions of imprinted genes, as one would expect based on the biological interpretation of MML and PDM.

We have corrected the issue with the P-values in accordance to our answer in question 9.

[1] M. S. Bartolomei and A. C. Ferguson-Smith. "Mammalian genomic imprinting." *Cold Spring Harbor Perspectives in Biology*. 3(7), a002592 (2011).

11. Page 20: I am curious about how the algorithm worked on the X chromosome in females? Or did you excluded this area from the analyses? I do not see this explicitly in your methods. Could you expand what will you expect on those areas?

The human subject on which we applied our analysis (STL003) is a male. Since males have only one copy of each sex chromosome, we only performed allele-specific methylation analysis on autosomes in our paper. However, the reviewer raises an interesting point about the possibility of using our analysis to detect allele specific methylation imbalances between the two copies of the X chromosome in a female subject, which is possible.

There are two types of X chromosome inactivation: imprinted and random [1]. The former is parent-specific and involves paternal silencing. The latter is not parent-specific and involves the random silencing of either copy. Only one copy remains active in both cases in order to avoid a potentially toxic double dose of X-linked genes [1]. Nevertheless, when considering a population of cells, the first type of inactivation will lead to each single cell in the population to have the same copy of the X chromosome inactivated, whereas this will not necessarily be the case in the second type of inactivation. As a result, the allele of origin might not contain information about the methylation state when inactivation happens randomly, and we do not expect to detect allele-specific methylation imbalances in this case.

When it comes to the first type of inactivation (i.e., imprinting), we expect to see a large number of MML-haps and PDM-haps, since X-chromosome inactivation works partly through DNA methylation silencing in the inactive copy [2]. This follows from the fact that we expect to see large mean methylation level differences between alleles, as well as large differences between the probability distributions of methylation, especially for genes fully expressed in the active copy and fully silenced in the inactive copy. On the other hand, we do not necessarily anticipate large DNA methylation entropy differences between alleles, given the almost binary-like behavior of X-chromosome inactivation.

Therefore, we expect to see clear allele-specific imbalances (MML/PDM-haps) when X chromosome inactivation happens in a parental specific way, but we do not expect that to be the case when inactivation is random.

[1] J. Ahn and J. Lee. "X chromosome: X inactivation." *Nature Education* 1(1), 24 (2008).

- [2] A. M. Cotton, E. M. Price, M. J. Jones, B. P. Balaton, M. S. Kobor, and C. J. Brown. "Landscape of DNA methylation on the X chromosome reflects CpG density, functional chromatin state and X-chromosome inactivation." *Human Molecular Genetics* 24(6), 1528-1539 (2015).

REVIEWER #3

1. *Are the SNPs included when profiling methylation state? If so, how does bismark-based strategy distinguish methylated C from unmethylated T allele?*

Heterozygous CpG sites, which are created or removed by SNPs, must be taken into account when performing allele-specific methylation analysis, a task that the CPEL method handles in a rigorous manner. For example, if ALLELE 1 of a haplotype contains CpG sites 1, 2, 3, 4, 5 (but not CpG site 6 due to a SNP) and its homologous ALLELE 2 contains CpG sites 1, 2, 4, 5, 6 (but not CpG site 3 due to a SNP), then the CPEL method computes a CPEL model [i.e., an Ising model given by Eqs. (1)-(3) in the Main Text] $p_1(x_1, x_2, x_3, x_4, x_5)$ for ALLELE 1 using all available WGBS data associated with ALLELE 1 and a CPEL model $p_2(x_1, x_2, x_4, x_5, x_6)$ for ALLELE 2 using all available WGBS data associated with ALLELE 2. Subsequently, and before carrying out differential analysis, the CPEL method computes probability distributions of methylation (PDMs) $p_1(x_1, x_2, x_4, x_5)$ and $p_2(x_1, x_2, x_4, x_5)$ for each allele defined only over the homozygous CpG sites 1, 2, 4, 5, by marginalizing the previous CPEL models over the heterozygous CpG sites 3 and 6. Then, allele-specific methylation analysis is performed using these PDMs (see Sections 7.2, 8.2, and 9.2 in the Supplementary Information document). Notably, appropriately handling heterozygous CpG sites is unique to the CPEL method and novel. We have added a short discussion on this issue in page 27 of the revised Main Text.

The Bismark-strategy followed by Arioc, the methylation caller we used in this paper, cannot distinguish an unmethylated C in one allele associated with a C-T SNP, which will be transformed to a T during bisulfite sequencing and PCR, from a T associated with the homologous allele. To avoid this and other potential confounding effects that introduce bias in the methylation calls at heterozygous CpG sites, we disregard the methylation calls performed by Arioc at these specific sites and treat these calls as missing data during model estimation via maximum likelihood.

2. *How would the proposed method scale when haplotype blocks are very large as in F1 hybrid mouse?*

There are two strategies to go about the analysis proposed by the reviewer.

The first strategy is to partition large haplotype blocks into smaller haplotypes and apply our method on the resulting partitions. This is the only strategy that can use the current version of our method to address the problem mentioned by the reviewer. However, this approach assumes that the methylation states within a haplotype partition are statistically independent from those in a nearest-neighbor partition, which will not be true in general. Therefore, statistical analysis results obtained using this strategy should not be fully trusted when divisions of haplotype blocks are frequent.

The second (desired) strategy would be to allow for more than one parameter β in the potential energy function of the CPEL model [i.e., the Ising model given by Eqs. (1)-(3) in the Main Text]

characterizing DNA methylation within the entire (very large) haplotype (e.g., each allelic subregion could be characterized by its own α and β parameters). Then, by following similar mathematical steps as the ones provided in the Supplementary Information document, we could obtain formulas for efficiently computing the partition function as well as the statistical summaries used in this paper. Regarding the hypothesis testing part, a permutation test might be more preferable in this case than the bootstrap approach currently employed by the CPEL method, whose intent is to analyze haplotypes of relatively small sizes. Note, however, that this strategy requires the derivation of new mathematical formulas and development of new software. Moreover, reliable parameter estimation in this case will require using a better DNA methylation sequencing technology than WGBS that provides higher coverage, much longer and accurate reads, as well as reduced rates of missing data. Unfortunately, such a technology does not currently exist, with a possible exception ONT sequencing, which currently cannot provide high enough base-call accuracy among other issues.

3. Please show the distribution of MML, NME, and PDM.

We have done as suggested. We now provide boxplot distributions of MML and NME values associated with the two homologous alleles of the haplotypes identified by the CPEL method in each tissue of our data in Supplementary Fig. 8. We also provide boxplot distributions of the values of the three test statistics T_{MML} , T_{NME} , and T_{PDM} associated with significant and nonsignificant haplotypes identified by the CPEL method in each tissue of our data in Supplementary Fig. 9. We have also added a brief discussion of these results on pages 19 of the Main Text.

4. Please show if CPEL recovered parameter values that are used in the simulation.

We performed additional analysis to address this issue and summarized our results in the (new) Supplementary Fig. 2, which shows that our method can almost always reliably recover the true parameter values used in the simulation. Note however that in those cases that this is not true, our method recovers a probability distribution of methylation (PDM) that is close to the true PDM, and this is the only thing that matters in our ASM analysis approach, since our analysis is based on estimated PDMs. We have added a new paragraph explaining this on pages 13-14 of the Main Text.

5. Is it feasible to run simulated annealing many times to see the stability of parameter estimation?

We did so in the original version of the paper, by performing parameter estimation 100 times using partially observed data, and discussed the results in Section 4 of the Supplementary Information document (Section 5 in the new version). Figure S1 in Supplementary Information shows that, when using simulated annealing with temperature reduction factor 10^{-4} (algorithm SA3), the median estimation error (quantified by the Euclidean distance of the estimated parameter values from their true values) is the smallest, and the same is true for the interquartile range (IQR). This, together with the parameter estimation results depicted in (the new) Supplementary Figure 2, which show that use of this optimization algorithm recovers the true parameter values with increasing accuracy as more data are available, demonstrates the stability of the SA3 algorithm when employed for parameter estimation. We added a comment to that effect at the end of Section 5 in the Supplementary Information document.

6. Clarify what “this important issue” is in Line 15 Page 10.

We have removed this statement.

7. Remove NPI, NPD, and CPEL from Figure 1 legend.

The labels the reviewer refers to are necessary to properly reference the models depicted in the figure. One objective of Figure 1 is to illustrate the differences in data usage and modeling between the three models (NPI, NPD, and CPEL) discussed in the paper and, therefore, the labels help to identify each model in the figure. Note that we have simplified the legend of this figure as per Reviewer's #1 request (see #3).

8. Define what A is in Equation (5).

We added the definition of A in the Main Text; see page 10 of the new version.

REVIEWER #4

- 1. CPEL uses a relatively complicated bootstrap process to obtain p-values for statistical test. It is unclear, however, whether such test is calibrated in terms of type I error control. Demonstrating the CPEL can provide well controlled type I error is particularly important here, given that the identified MML-haps by CPEL are somehow highly enriched in CpG islands (CGIs), which contradicts almost all previous literature. Therefore, it would be important to show that CPEL controls type I error well both in simulations and in real data applications. It would be useful to simulate the null data under the Ising model, and then perform a similar bootstrap procedure as used in the real data, to examine whether the type I error is well controlled in CPEL (as well as in NPI and NPD). It would be important to look at type I error control at the genome-wide significance level used in the real data applications.**

We agree with the reviewer that showing good control of Type I error is important. Therefore, we followed this suggestion and performed new simulations from which we computed empirical estimates of the cumulative distribution functions of the 'null' P -values that turned-out to be linear (see our new discussion on pages 14-16 in the Main Text, and the new Supplementary Fig. 3). This shows that the 'null' P -values follow a uniform distribution, which implies that CPEL's hypothesis testing method will provide proper control of the Type I error under the null hypothesis, since the probability of obtaining a P -value that is below a significance level α will precisely be α in this case, implying a Type I error of $\alpha\%$. Analysis of our real data leads to the same conclusion (see our new discussion on page 18 in the Main Text and the new Supplementary Figs. 5 & 6a). In addition to the above, and by following the reviewer's suggestion, we looked at the Type I error after correcting for multi-hypothesis testing using the Benjamini-Hochberg procedure and consistently found that this error is zero (see our new discussion on page in the Main Text and the new Supplementary Fig. 6b). We believe that these results should convince the reader that the CPEL method performs hypothesis testing in a statistically sound manner.

Demonstrating the same for the NPI and NPD methods is not useful, since these methods do not test for NME and PDM imbalances as the CPEL method does. Moreover, we have clearly shown that the NPI and NPD methods are fundamentally problematic for hap-ASM analysis. Therefore, we do not see the point for studying hypothesis testing using these methods.

Finally, we would like to point out that, in addition to CpG islands, the MML-haps detected by the CPEL method were highly enriched in promoter regions of known imprinted genes. This suggests

that the detected MML-haps are true positives, since we expect stark imbalances in mean methylation level in this case. In addition, the PDM-haps detected by the CPEL method were also highly enriched in promoter regions of imprinted genes, which is consistent with our expectation of observing substantially different behaviors in stochastic methylation within two homologous alleles, resulting in significant imbalances in the probability distribution of methylation. Moreover, promoter regions of imprinted genes were not enriched in NME-haps detected by the CPEL method. This is consistent with the fact that imprinting is associated with two ordered states, a fully methylated and a fully unmethylated state, each associated with zero entropy and, therefore, we do not expect imbalances in methylation entropy between the two alleles in this case. These results serve as further evidence that the CPEL method detects true and important methylation imbalances.

- 2. It would be important to provide the $-\log_{10}$ p-value distribution in the real data through qq-plots, for each of the three tests (MML, NME, PDM) and in each tissue. Because the majority of the haplotypes do not show significance in the real data and are effectively null, one would expect the small $-\log_{10}$ p-values to adhere on the diagonal line on the qq-plot.***

We did as the reviewer suggested (see new discussion on page 18 in the Main Text and the new Supplementary Fig. 7). As the reviewer suggests, large P -values associated with haplotypes that do not exhibit significant methylation imbalances in terms of the three test statistics in Eq. (5) of the Main Text must behave as expected under the null hypothesis (i.e., they should be samples drawn from a uniform distribution). In this case, small observed $-\log_{10} P$ -values must adhere to the diagonal of a Q-Q plot of observed vs. expected quantiles, which is shown to be true in our data (see new Supplementary Fig. 7). Note, however, that although some small $-\log_{10} P$ -values are located on the diagonal line in each plot, other small values are located slightly below this line. This is due to the conservative nature of the hypothesis testing approach used by the CPEL method, which results in slight overestimation of the true P -values; see Section 10 of the Supplementary Information document. This is a consequence of the fact that the CPEL method generates ‘null’ statistics at the minimum coverage and the most complex model observed in the data. As a result, the ‘null’ statistics will contain the largest possible amount of observed statistical variability, resulting in a rather conservative test and ensuring proper control of Type I error under all circumstances.

- 3. It would be important to report the genomic control factors for each test and in each tissue. A genomic control factor that is close to 1 would indicate reasonably good type I error control in the real data analysis.***

Based on our (limited) understanding, genomic control factors are employed in genome-wide association studies (GWAS) in which hypothesis testing is performed for detecting genetic markers using data from multiple subjects. For a genetic marker, a statistic is computed using data from multiple individuals from which a P -value is calculated for evaluating statistical significance. However, observed statistics in association studies can be confounded by the presence of subgroups in a population with ancestry differences. Consequently, neglecting or not accounting for these differences among sample individuals can lead to a high Type I error or spurious associations, which is a serious concern.

Our method was not conceived to be a tool for population-level studies, since allele-specific methylation analysis is typically performed independently for each subject and tissue. In sharp contrast to GWAS, the CPEL method independently performs hypothesis testing at each

haplotype using data from a single subject and a single tissue (see Section 10 of the Supplementary Information document). That is, for a given subject and tissue, a statistic is computed at each haplotype and a P -value is subsequently calculated from this statistic. Since, in this case, the statistic used for hypothesis testing does not use data from multiple individuals, there is no need to account for ancestry differences among sample individuals. It therefore seems to us that using genomic control factors in the current framework is not appropriate. In fact, genomic control factors have not been used in allele-specific methylation studies to the best of our knowledge. Nonetheless, the evidence provided in our answers to the previous two remarks show clearly, by using both simulations and real data, that the CPEL method is capable of properly controlling the Type I error thanks to the conservative nature of the hypothesis testing scheme used.

- 4. Some part of the bootstrap procedure is not realistic and can be improved. For example, right now the reads are placed into two complementary groups that contain the same number of reads. The constraint that the two groups contain the same number of reads seems rather unrealistic. A better choice would be to randomly assign reads onto these two group without such equal read constraint. This can be easily achieved by assigning each read onto two haplotypes through an over-dispersed binomial distribution (beta-binomial) with mean of 0.5. This more realistic strategy could help improve type I error control and reduce potential false discoveries.***

The reviewer raises an interesting point. However, we believe that implementing his suggestion would in fact result in worse Type I error control and a potential increase in false discoveries. Although we have argued this issue in Section 10 of the Supplementary Information document, we also include here a brief discussion (see also our response to comment #2 above).

Since our objective is to control the Type I error, we choose to assign the minimum possible coverage to each allele, which requires that both alleles have the same coverage. This strategy maximizes statistical variability in the parameter estimates of both CPEL models and, therefore, the computed null statistics will be subject to maximum statistical variability as well. This ensures that the method does not compute observed test statistic values at a lower coverage than the coverage used to compute null statistics and, hence, P -values. As a result, this strategy will slightly overestimate P -values resulting in conservative hypothesis testing.

By doing what the reviewer proposes would reduce the amount of statistical variability in one of the two alleles. This could result in a less conservative empirical null distribution and, therefore, reduced control of the Type I error. For this reason, we think that by equally splitting the reads is the best way to ensure that the Type I error is properly controlled, albeit at the cost of some miss-detections.

- 5. A key potential bias for allelic specific methylation analysis is due to read mapping. Specifically, when a CpG site is in a neighborhood of a SNP, it is much harder to map a methylated read to the alternative allele than an unmethylated read. Consequently, methylation level for the given CpG site maybe artificially higher on the haplotype with the reference SNP, leading to methylation bias towards the reference allele. This could cause potentially false signals in the NME analysis, and, likely to a lesser extent, in the MML analysis. If you plot methylation levels of the CpG sites on the alternative SNP allele vs the reference SNP allele, would you see a bias? If there is a mapping bias, does the mapping bias occur preferentially on CGIs (or other regions with high CpG density), so that it creates a false enrichment of MML-haps on CGIs? The mapping bias can be mitigated, though not completely eliminated, by relaxing the mismatches during the reads mapping step. The***

mapping bias can also be mitigated by excluding CpG sites with an obvious difference in methylation levels between reference and alternate alleles. I am a bit concerned whether such mapping bias may introduce false signals in the MML and NME analysis.

We mask the reference genome prior to performing alignment using SNPsplit. This step replaces all SNP positions by 'N' nucleotides using the SNP information available in the VCF file of subject STL003. As a result, there is no penalization in the mapping score for either allele due to a potential mismatch in the SNP position between the read and the reference genomes (we added this comment in the new version of our paper; see page 30). In addition, the aligner we used is designed specifically for WGBS data, and ensures that there is no bias during the mapping of WGBS reads due to methylation levels (see [1] and [2] below). From [1], "Since residual cytosines in the sequencing read are converted *in silico* into a fully bisulfite-converted form before the alignment takes place, mapping performed in this manner handles partial methylation accurately and in an unbiased manner." Thus, there is no bias in the mapping of WGBS reads. In fact, as our Supplementary Table 1 shows, the average coverage of both alleles is the same in all tissues.

[1] F. Krueger and S. R. Andrews. "Bismark: a flexible aligner and methylation caller for Bisulfite-Seq applications." *Bioinformatics* 27(11), 1571-1572 (2011).

[2] R. Wilton, X. Li, A. P. Feinberg, and A. S. Szalay "Arioc: GPU-accelerated alignment of short bisulfite-treated reads." *Bioinformatics* 34(15), 2673-2675 (2018).

6. *It would be important to apply NPI and NPD to all enrichment analysis in the real data and compare their results with CPEL. It is equally important to examine the ability of NPI and NPD in terms of identifying significant regions overlapping the promoter regions of imprinted genes, across different tissues. Comparing CPEL with NPI and NPD on imprinted genes would provide a much stronger evidence for the statistical power difference between CPEL and the other two methods in the real data applications.*

It is not possible to reliably apply the NPD method on our real data and, therefore, we cannot repeat our enrichment analyses using this method in order to compare it with the CPEL method. As we have explained in our paper, the NPD method requires estimation of 2^N probabilities within an allele containing N CpG sites, which is not possible in general due to lack of sufficient WGBS data required for reliable estimation. Our simulation results, depicted in Fig. 3 of the Main Text and in Supplementary Fig. 1, show clearly how bad NPD model estimation can be, even at high coverage and small number of CpG sites. Moreover, we have extensively discussed this issue in Section 3 of the Supplementary Note in [1].

Unfortunately, we cannot compare all enrichment analysis results obtained by the CPEL method to ones obtained by using the NPI method (or any other method), since there is no known ground-truth, which will make any attempted comparison questionable at best. But even if this issue is ignored, there is strong statistical evidence that the CPEL method is superior to the NPI method in many respects. First, the CPEL method takes into account correlations in DNA methylation, whereas the NPI method does not. Second, NPI model estimation is not as reliable as CPEL model estimation under various circumstances (see Fig. 3 in Main Text and Supplementary Fig. 1). Third, analysis of our real data using Akaike's Information Criterion (AIC) shows that the CPEL model is preferred over the NPI model, especially within alleles containing at least 4 CpG sites (see Fig. 4 in Main Text). This

shows that the CPEL model provides a better trade-off between ‘goodness of fit’ to available data and model simplicity (in terms of the number of parameters used) than the NPI model.

Examining the ability of the NPI method to identify significant haplotypes overlapping the promoter regions of imprinted genes across different tissues and comparing that to CPEL’s seems an attractive idea, since in this case there is some ‘ground-truth’ information to work with. However, we have shown in [2] that the unrealistic assumption of statistical independence may lead to loss of specificity (true negative rate) and sensitivity (true positive rate). This will seriously affect the statistical performance of a method based on statistical independence, such as the NPI method (see our remark on page 4 of the Main Text and on page 23 of the Supplementary Information document). In addition, such a method may lead to sensitivity (true positive rate) that equals the Type I error rate (false positive rate), indicating a performance that is no better than random guessing [2] (see also our remark on page 23 of the Supplementary Information document), which implies that the NPI method cannot always control the Type I error. Therefore, it does not make sense to compare the statistical power difference between the COEL and NPI methods, considering the fact that only the CPEL method always controls the Type I error, as we have shown in the new version of our paper.

[1] G. Jenkinson, E. Pujadas, J. Goutsias, and A. P. Feinberg, “Potential energy landscapes identify the information-theoretic nature of the epigenome.” *Nature Genetics* 49(5), 719-729 (2017).

[2] G. Jenkinson, J. Abante, A. P. Feinberg, and J. Goutsias. “An information-theoretic approach to the modeling and analysis of whole-genome bisulfite sequencing data.” *BMC Bioinformatics* 19, 87 (2018).

7. ***CPEL is effectively a simplified version of their previous method (introduced in ref #10). The simplifications include (a) removing the dependence of alpha on CpG density; (b) remove the dependence of beta on distance between pairs of CpG sites. These simplifications seem counter-intuitive, given that both features (a) and (b) are rather desirable for real data applications. Therefore, it would be important to compare CPEL with their previous method in all real data applications to demonstrate the benefits of removing features (a) and (b).***

To address this comment, we have added a detailed explanation in the Supplementary Information document (see new Section 2). Briefly, we cannot apply our previous method in ref #10 for haplotype-dependent ASM analysis, since we need in this case to use probability distributions over the actual space of methylation patterns within relatively large alleles. This leads to a computationally expensive approach whose implementation takes an unrealistic amount of time, even on a high performance computer cluster. Note that, in order to address this issue in ref #10, we performed methylation analysis using probability distributions of methylation levels (and not methylation patterns) within small genomic units of 150 bp each, an approach that is not useful for haplotype-dependent ASM analysis.

As we explain in the Supplementary Information document, due to limitations in read-based phasing (often constrained by the length of WGS reads) and the low coverage of WGBS, current sequencing technologies allow analysis only of relatively small haplotypes (for example, the size of more than 99% of the haplotypes analyzed in our data is no more than 1-kb). Given the small amount of available WGBS data, it is reasonable to assume that we can accurately observe only the average of

the methylation means at individual CpG sites within small subregions of a given allele. Moreover, and since pair-wise correlations are second-order moments requiring more data than means for reliable estimation, we can also assume that we can accurately observe only the average of these correlations at individual CpG sites within the entire allele. For these reasons, we subdivide in the current work each allele into a minimum number K of equally sized non-overlapping subregions of size no more than G . Given the previous constraints, and by invoking the maximum-entropy principle, we can then show that the probability distribution of methylation patterns within an allele is associated with the potential energy function given by Eq. (2) in the Main Text. This energy function allows us to perform haplotype-dependent ASM analysis by a computationally feasible method based on probability distributions of methylation patterns instead of probability distributions of methylation levels, as it was the case in ref #10. As we explain in the Supplementary Information document, computations in this case can be performed efficiently by multiplying 2×2 matrices, which are evaluated by spectral decompositions that require eigenvalues and eigenvectors given by analytical formulas, as well as by employing standard derivative approximations and a limited number of Monte Carlo estimations. Consequently, we can perform haplotype-dependent ASM analysis in the original space of individual methylation patterns, which allows the CPEL method to identify types of significant allele-specific methylation imbalances completely inaccessible to currently available methods.

- 8. A critical parameter in the CPEL method is the subregion size, denoted by G . G was set to be 500bp. How stable are the real data results with respect to the choice of G ? Would these real data results change if you use a smaller G or a larger G ?**

Results would slightly change, since the value of G determines the scale at which the ASM analysis is performed and, in turn, the complexity of the models used. However, a compromise must be reached in order to balance model complexity and data availability, as it is often the case is model estimation.

Note that G directly affects the granularity of modeling. This is due to the fact that the methylation state within an allele of interest is associated with a potential energy function that is characterized by the $K + 1$ parameters $\alpha_1, \alpha_2, \dots, \alpha_K$ and β , where K increases with decreasing G . Consequently, a smaller value of G results in a finer description of the methylation state in terms of an increased number of model parameters. This however leads to the CPEL method being able to analyze a decreased number of haplotypes for a given WGBS coverage, since successfully estimating the values of an increasing number of parameters in each homologous allele requires higher coverage.

To determine an appropriate value for G that strikes a balance between a finer model description and the number of haplotypes that the CPEL method can analyze, we recommend using the scheme described on page 6 in the (new) Supplementary Information document. This scheme determines the smallest possible value of G (i.e., it provides the finest model description of the methylation state) that leads to no more than a 5% loss in the number of haplotypes that the CPEL method can analyze in given data when comparing to the case of maximum granularity. By using this strategy, we consistently found that $G = 500$ bp in all tissues in our data.

- 9. Is there a way to quantify whether these detected allelic specific methylation events are due to cis genetic differences? For example, do you tend to detect haplotypes with a high number of SNPs, a high density of SNPs, or a long region length?**

We are currently working on a follow-up paper in which we are using all data available from the Roadmap Epigenetics Project to study allele-specific methylation in detail and from a biological perspective. The sample size of this data set (48 samples) gives a significantly larger statistical power to detect interesting associations, such as the ones suggested by the reviewer. However, we refrained from including further biological results in the present paper since the main purpose of this paper is to present our novel method and study some of its properties and potential by using simulations and some real data analysis. In addition, our objective has been to demonstrate that our method is superior to the state-of-the-art and is capable of detecting phenomena inaccessible to the other methods. We view the present paper as a Methodology paper.

10. *In the real data applications, do detected regions of either MML, NME, or PDM correlated with TAD boundaries?*

Although the reviewer raises a very interesting point here, we will refrain from including more biological results in this paper as noted in our response to the previous question. We plan to investigate the reviewer's suggestion in a follow-up paper.

11. *It would be useful to report the computation time of CPEL in real data applications.*

We have added a comment on page 28 in the new version of the Main Text about this issue. It took the CPEL method about 48 hours using 20 CPUs to process one WGBS sample in our real data, which necessitates the use of a computer cluster when employing the CPEL method for hap-ASM. This computational effort is much less than the one required by informME, our previous method for methylation analysis [1]. Given that DNA sequencing is expensive and time consuming and taking into account the scale of ASM analysis, the complexity of the data required to perform such analysis, and the advantages that the CPEL method offers over existing methods (e.g., ASM analysis at the haplotype level, entropy computations, detection of statistically significant imbalances using rigorous hypothesis testing, and reliable and reproducible statistical analysis), we believe that the added computational cost should not be an issue. This is on par with other bioinformatics tools used in the particular problem at hand for WGS/WGBS data preprocessing and alignment, should not be an issue.

[1] G. Jenkinson, E. Pujadas, J. Goutsias, and A. P. Feinberg, "Potential energy landscapes identify the information-theoretic nature of the epigenome." *Nature Genetics* 49(5), 719-729 (2017).

12. *The bootstrap procedure in CPEL only keeps bootstrapped sites where CPEL estimation is successful. What's the success rates of this step in the bootstrap in real data applications?*

The success rate varies substantially depending on the number N of CpG sites for which the null statistics are being generated, the specific number K of subregions used, as well as the available coverage. However, this does not matter in our method, because CPEL performs hypothesis testing within a given haplotype only when a minimum number of required null statistic values are available (which we take it to be 1,000). We have discussed this in Section 10 of the Supplementary Information document (see Step 7 of the algorithm).

REVIEWERS' COMMENTS:

Reviewer #1 (Remarks to the Author):

The manuscript by Abente et al. entitled "Detection of haplotype-dependent allele-specific DNA methylation in WGBS data" explores how to convey both WGS and WGBS to detect allele specific DNA methylation.

The authors have addressed all the comments. I have a minor comment for the package as I was not able to install it, and this could be important for the release for the general public.

Minor comment:

The provided zip file (both as supplementary material and in your website) opens in this new version. I can see the contents and the files required for the test to run. However, when doing the local installation using Julia, the process crashed and I decided not to troubleshoot it more after several tries and errors. Due to this I can not provide any input. The README file is designed based on the github installation, so again I cannot tell whether this is a language dependent problem that I am unfamiliar or not. Please add options for a local installation in your README file for your users in case they do not have internet access. This is important during package development, in particular when you are using languages that are not as widely known for all your potential users. This is something that you could try to address for your release version.

A different option is to use a Docker function to embed your Julia version and package pre-compiled for the user. Again this is just a minor suggestion for the authors, beyond the quality of the manuscript.

Reviewer #3 (Remarks to the Author):

All my previous comments have been properly addressed, and I have no other issues.

Reviewer #4 (Remarks to the Author):

Most of my comments are well addressed. However, two of my major comments, #9 and #10, questioning the potential biological insights one can get by using the new method, are not addressed. Showing that a method is capable of discovering new biology is critical for demonstrating the practical utility of the method and thus is an essential component of a Methodological paper, especially for a high-quality biology journal.

REVIEWERS' COMMENTS:

Reviewer #1 (Remarks to the Author):

The manuscript by Abente et al. entitled "Detection of haplotype-dependent allele-specific DNA methylation in WGBS data" explores how to convey both WGS and WGBS to detect allele specific DNA methylation.

The authors have addressed all the comments. I have a minor comment for the package as I was not able to install it, and this could be important for the release for the general public.

Minor comment:

The provided zip file (both as supplementary material and in your website) opens in this new version. I can see the contents and the files required for the test to run. However, when doing the local installation using Julia, the process crashed and I decided not to troubleshoot it more after several tries and errors. Due to this I can not provide any input. The README file is designed based on the github installation, so again I cannot tell whether this is a language dependent problem that I am unfamiliar or not. Please add options for a local installation in your README file for your users in case they do not have internet access. This is important during package development, in particular when you are using languages that are not as widely known for all your potential users. This is something that you could try to address for your release version.

A different option is to use a Docker function to embed your Julia version and package pre-compiled for the user. Again, this is just a minor suggestion for the authors, beyond the quality of the manuscript.

We did extensive testing in multiple computers with various operating systems to ensure that both the local installation and the toy example run flawlessly. In addition, we included a README file describing the required steps for the entire process. Although we are very sorry to hear that the reviewer could not install the package and try it, we believe that this could be related to external factors unrelated to the package itself.

Following the reviewer's suggestion, we added the steps for the local installation of the package, as well as the steps for the toy example, to the Github README file. Nevertheless, the published release (v0.0.1) can be flawlessly installed by using a single command in Julia, which significantly simplifies the entire installation process. At this point, this is the main way in which users will be installing the package. In addition, the repository is setup such that the code is automatically tested in the three major operating systems (Windows, macOS, Unix) after each update, ensuring that the latest available version works in all cases.

Reviewer #3 (Remarks to the Author):

All my previous comments have been properly addressed, and I have no other issues.

Reviewer #4 (Remarks to the Author):

Most of my comments are well addressed. However, two of my major comments, #9 and #10, questioning the potential biological insights one can get by using the new method, are not addressed. Showing that a method is capable of discovering new biology is critical for demonstrating the practical utility of the method and thus is an essential component of a Methodological paper, especially for a high-quality biology journal.

We agree with the reviewer that showing new biology is important for demonstrating the practical utility of the method. However, we also think that it is important to show that the results produced by the method are consistent with previous results when available. For instance, we show how our method produces results that are remarkably consistent with previously reported imprinted genes. Furthermore, we provide several new results, such as the fact that observed co-occurrence of NME-haps across different tissues is higher than that of MML-haps. This difference in prevalence between the two types of haps suggests that methylation entropy differences could be more sequence driven than mean methylation level differences. Another example is the depletion of NME-haps observed in CG rich regions and the enrichment of such haps in CG poor regions. This result is consistent with the common understanding that DNA methylation is more regulated in high CpG density regions since these regions often coincide with promoter regions. These are just two of many 'biological' insights that we obtained using our method on a subset of the dataset generated in Onuchic et al. 2018.

Nevertheless, although the reviewer is asking two interesting biological questions, we want to stress that this paper's focus is the method itself and the release of user-friendly software for the community. However, as we have shown in our current paper, our method allows for the study of allele-specific methylation differences in a much more comprehensive manner compared to the previous work. Thus, in addition to the results reported in our current paper, we believe our method will provide valuable biological insights in future studies.

Lastly, as we pointed out in our previous response, we are currently performing an extensive biological analysis using the entire dataset from Onuchic et al. 2018. The two biological questions suggested by the reviewer are among the many biological questions that we are addressing in this new analysis. Adding these results and all associated explanations and discussions to the current paper will result in a complex paper whose size could be twice the size of the current paper.